# Purely self-rectifying memristor-based passive crossbar array for artificial neural network accelerators

Kanghyeok Jeon[1,2,7], Jin Joo Ryu[2,3,7], Seongil Im[4], Hyun Kyu Seo[5], Taeyong Eom[2], Hyunsu Ju[4] ✉, Min Kyu Yang[5] ✉, Doo Seok Jeong [1] ✉ & Gun Hwan Kim [3,6] ✉

Memristor-integrated passive crossbar arrays (CAs) could potentially accelerate neural network (NN) computations, but studies on these devices are limited to software-based simulations owing to their poor reliability. Herein, we propose a self-rectifying memristor-based 1 kb CA as a hardware accelerator for NN computations. We conducted fully hardware-based single-layer NN classification tasks involving the Modified National Institute of Standards and Technology database using the developed passive CA, and achieved 100% classification accuracy for 1500 test sets. We also investigated the influences of the defect-tolerance capability of the CA, impact of the conductance range of the integrated memristors, and presence or absence of selection functionality in the integrated memristors on the image classification tasks. We offer valuable insights into the behavior and performance of CA devices under various conditions and provide evidence of the practicality of memristor-integrated passive CAs as hardware accelerators for NN applications.

Artificial neural networks (ANN) are indispensable for a wide range of artificial intelligence (AI) applications, including real-world data processing, such as pattern recognition, classification, and predictive modeling[1–8]. However, the growing complexity of these applications requires advanced ANN architectures capable of processing vast amounts of data with high precision and low power consumption. The simultaneous development of software and hardware is crucial for accelerating NN computations. Various software-level approaches have been proposed to obtain lightweight semantic segmentation networks, including quantization, compression, and lightweight architecture design[9–14]. Quantization and compression are effective strategies, but result in a large loss of accuracy[9–12]. By contrast, the design of lightweight architecture, such as depth-wise separable convolution, can enhance computational efficiency without sacrificing accuracy[13,14]. Hardware accelerators such as graphic processing units

(GPUs), field programmable gate arrays (FPGAs), and application-specific integrated circuits (ASICs) can speed up NN algorithms[15–18]. However, these devices are still limited by the memory wall in the von Neumann architecture and cannot meet the high-speed and low-energy-consumption requirements of ANNs.

The process-in-memory (PIM) computing architectures have recently been proposed to overcome the limitations of conventional systems for advanced ANNs. These architectures have gained popularity as alternatives to conventional computing systems because of their highly advantageous system topology, which enables the concurrent implementation of data processing and storage in a single chip. When processing units are directly integrated into the memory of a PIM architecture, computations can be performed in the memory, thereby eliminating the need for data transfer between the memory and processing units, significantly reducing data movement overheads

[1]Division of Materials Science and Engineering, Hanyang University, Seoul 04763, Republic of Korea. [2]Division of Advanced Materials, Korea Research Institute of Chemical Technology (KRICT), Daejeon 34114, Republic of Korea. [3]Department of Materials Science and Engineering, Yonsei University, Seoul 03722, Republic of Korea. [4]Center for Opto-Electronic Materials and Devices, Korea Institute of Science and Technology (KIST), Seoul 02792, Republic of Korea. [5]Intelligent Electronic Device Lab, Sahmyook University, 815 Hwarang-ro, Nowon-Gu, Seoul 01795, Republic of Korea. [6]Department of System Semiconductor Engineering, Yonsei University, Seoul 03722, Republic of Korea. [7]These authors contributed equally: Kanghyeok Jeon, Jin Joo Ryu. ✉e-mail: hyunsuju@kist.re.kr; dbrophd@syu.ac.kr; dooseokj@hanyang.ac.kr; kgh@yonsei.ac.kr

and enhancing energy efficiency. Moreover, the PIM strategy enables parallel data processing with reduced energy consumption and increased memory bandwidth, rendering it a promising solution for applications that require high speed, low power consumption, and complex ANN operations[19–23]. Indeed, PIM computing has achieved 10- and 100–1000-fold improvements over CPU and GPU accelerators in terms of speed and energy efficiency, respectively[24].

Memristive crossbar arrays (CAs) have emerged as a key component that serves as the memory unit in PIM architectures. These CAs enable fundamental computational operations, with vector matrix multiplication (VMM) as a crucial operation that exploits the principles of Ohm's and Kirchhoff's laws. Accelerating VMM operations using memristive CAs offers significant advantages in tasks that involve heavy matrix computations such as speech and image classification. The processing efficiency of these matrix-intensive tasks can be significantly enhanced by harnessing the power of memristors. A key benefit of memristors is their non-volatile nature, which profoundly impacts the overall computing system performance. The non-volatility of memristors eliminates the need for frequent data fetching and communication between the memory and processing units. This reduction in data movement minimizes latency and results in substantial energy savings. Consequently, memristor-based CAs could potentially improve the performance of computing systems, particularly in applications in which speed and energy efficiency are critical.

Recent studies have utilized memristor CAs to implement NN applications. Various memristor devices, including magnetic random-access memory (MRAM), phase-change random-access memory (PRAM), and resistive random-access memory (RRAM), have been integrated into CAs to accelerate NN computations. In particular, RRAM has gained significant attention owing to its favorable characteristics, including high scalability, good analog-switching ability, excellent endurance, low power consumption, and high switching speed[25–35].

However, the integration of memristors into CAs is critically challenged by sneak currents from neighboring cells, which can lead to interference and inaccurate results. This issue is commonly addressed by inserting selection functionality into CAs. Various approaches, including the integration of transistors with memristors in a one-transistor one-memristor (1T1M) configureation[31,36–39], and the incorporating of selector devices such as ovonic threshold switching[40–42], mixed ionic-electronic conductors[43], and field-assisted super-linear threshold switching devices[44] in a one-selector one-memristor (1S1M) configuration have been explored.

However, integrating transistors in the 1T1M configuration can result in area overhead issues, and integrating selectors in the 1S1M configuration poses practical and compatibility challenges for memristors. Self-rectifying memristors (SRMs) have emerged as a promising solution to overcome these challenges. They possess inherent selection functionality, which allows them to effectively suppress sneak currents and enable accurate and efficient VMM operations in CAs. The need for additional transistors or selectors can be eliminated by leveraging the inherent rectifying behavior of SRMs, leading to fewer area overhead and compatibility issues and straightforward and efficient VMM operations.

Most previous research implementing NN applications, particularly classification, has concentrated on simulating the performance and accuracy of CA computations, rather than showcasing practical computations within memristor-based CAs. Hardware-based implementations are highly practical and straightforward; however, the resolution of reliability issues associated with the CA and integrated memristors that constitute the hardware accelerator is challenging. Several conditions must be satisfied to achieve the practical implementation of this approach. First, the CA must have sufficient yield and functional control. Second, the memristor device must exhibit reliability in terms of non-volatility, uniform operating characteristics across all cells, selection ability, and operational repeatability. Developing a hardware-accelerating system that fulfils these conditions is extremely difficult, which explains the limited number of research results on hardware-based implementations for NN applications. Recent studies have shown that memristor-based NN computations can accelerate VMM operations for real-world data classification such as facial images and the Modified National Institute of Standards and Technology (MNIST) dataset. Hu et al.[36]. utilized a 128 × 64 memristor-based CA for MNIST dataset classification and achieved 89.9% classification accuracy using a 1T1M-based CA to prevent sneak currents. Kim et al.[45]. integrated TiN/Al/Ti/TiO$_{2-x}$/Al$_2$O$_3$/TiN stacked memristors into a 64 × 64 passive CA and demonstrated high recognition accuracy for MNIST image classification. However, no studies on fully hardware-based image classification implementations using SRM-based passive CAs have been demonstrated. Moreover, most studies on SRMs have not addressed reliability issues, such as non-volatility, lack of sufficient selectivity, and feasibility of pulse-based operations, which are crucial for practical CA circuit operations. We had previously demonstrated.[46] a highly reliable and energy-efficient SRM-based passive CA with excellent performance, and identified the feasibility

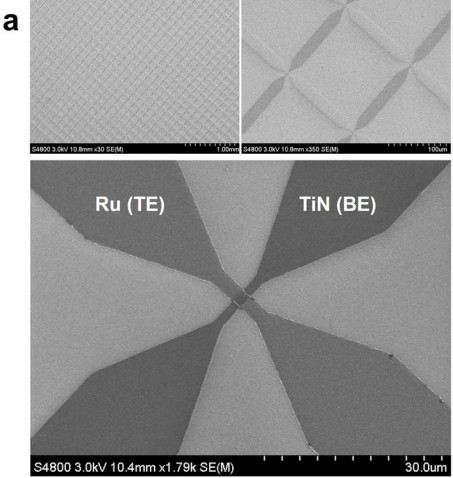
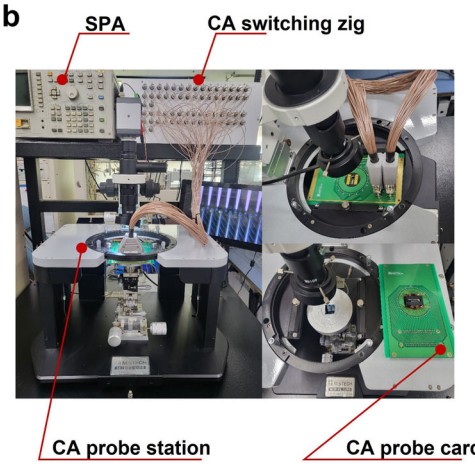

**Fig. 1 | CA device images and equipment for electrical performance. a** SEM images of the 1 kb CA device. **b** Photographs of the electrical equipment used to characterize the CA devices. The equipment included a semiconductor parameter analyzer (SPA), CA probe station, CA switching zig, and CA probe card.

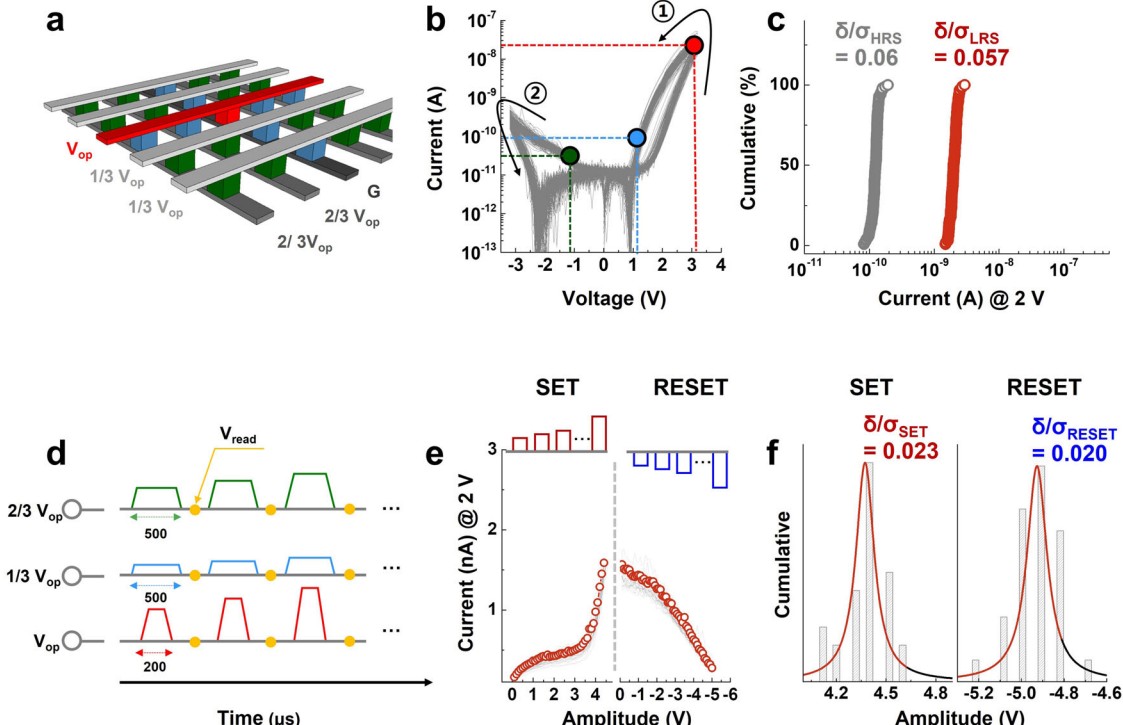

**Fig. 2 | Electrical characteristics of our 1 kb CA device. a** Schematic of the CA utilizing a one-third biasing scheme during the DC $I–V$ characterization. **b** DC $I–V$ characteristics of 1024 cells in the CA. The black curve marked ① denotes the SET process, while the black curve marked ② denotes the RESET process. **c** Reading current distributions of each state. The coefficient of variations of the LRS and HRS were 0.057 and 0.06, respectively. **d** Schematic of the AC-based operations utilizing the biasing scheme. Inhibiting voltages were induced in the unselected cells and had a larger pulse width than the operational voltage. **e** Electrical pulse-induced behavior of 30 cells in the CA. **f** Amplitude distributions for the SET and RESET processes. Very low deviation values were observed in each process. The highly uniform electrical characteristics of the CA can be attributed to the nonfilamentary characteristics of our previously developed SRM device.

of a VMM accelerator using this CA; however, the device did not achieve actual data processing, such as image classification.

In this study, we propose a fully hardware-based ANN computation accelerator using a 1 kb passive CA integrated with a previously developed HfSiO$_x$-based SRM. We successfully fabricated the 1 kb CA in 100% yield. The high selectivity, low device-to-device (D2D) variation, and robust non-volatility of the developed device enabled highly reliable VMM operations. Training and inferencing for image classification operations were also achieved with very low error rates. We then conducted experiments to validate the impact of key variables on the inferencing accuracy of an SRM-based hardware accelerator. First, we investigated the effect of the inclusion of 30% and 50% defective cells in the 1 kb CA on the inferencing accuracy in image classification operations. Second, we explored the impact of the memristor's reading margin (on/off ratio) on the classification accuracy. Although previous researchers assumed that a large reading margin is essential to improve inferencing ability, we found that the reading margin of the memristive device does not significantly affect its inferencing accuracy. Finally, we emphasized the importance of incorporating selection functionality into CAs for the reliable operation of ANN computation accelerators. We comparatively investigated the impact of the absence of selection functionality on the accuracy of VMM operations by integrating a memristive device without selection functionality into an 8 × 8 CA. We found that selection functionality is essential for accurate VMM operations in a CA, and that the absence of selection functionality causes a significant degradation in NN computing accuracy.

## Results
### Electrical characteristics of the SRM-based 1 kb passive CA
A previously developed HfSiO$_x$-based SRM was integrated into a 1 kb CA (32 × 32). Figure 1a shows the scanning electron microscopy images

of the CA. The CA was constructed using a straightforward structure in which the patterned bottom (BE) and top (TE) electrode layers were connected perpendicularly via a crosspoint and individual SRMs were formed at each junction. The line width for each line comprising the top and bottom electrodes is set at 2 μm, resulting in an effective junction area of 4 μm². The fabrication details of the CA devices are described in the Methods section. The electrical characterization equipment is shown in Fig. 1b. A customized 32 × 32 CA switching zig, CA probe station, and probe card were used for the individual-cell-accessible characterization of the CA.

Figure 2a shows a schematic of the CA and the applied biasing scheme used for its electrical operation. The red line represents the biasing line for the operational voltage ($V_{op}$), and the dark-gray line represents the ground (G). To prevent sneak currents from neighboring cells, we biased the other upper and lower lines that do not share the accessed cell (red cell) to 1/3 $V_{op}$ (light gray) and 2/3 $V_{op}$ (gray), respectively; these voltages are referred to as inhibiting voltages. Under this biasing scheme, if the voltage-drop segment through the metal line is ignored (the resistance of SRM is much higher than that of metal line.), the green and blue cells in the CA are biased to −1/3 $V_{op}$ and 1/3 $V_{op}$, respectively. Thus, $V_{op}$ is applied to the accessed cell only. In a previous report, we investigated various biasing schemes to suppress sneak currents in the CA using an identical SRM[46,47]. We observed negligible differences between the one-half and one-third biasing schemes because of the sufficiently high selectivity of our SRM (Supplementary Fig. S1). Furthermore, the numerical analyses for the limited density of the array and additional power consumption stem from the sneak current under applying each biasing scheme were conducted (Supplementary Fig. S2).

Using the biasing scheme described above, we examined the DC current–voltage ($I–V$) characteristics of all cells (1024 cells in total) in

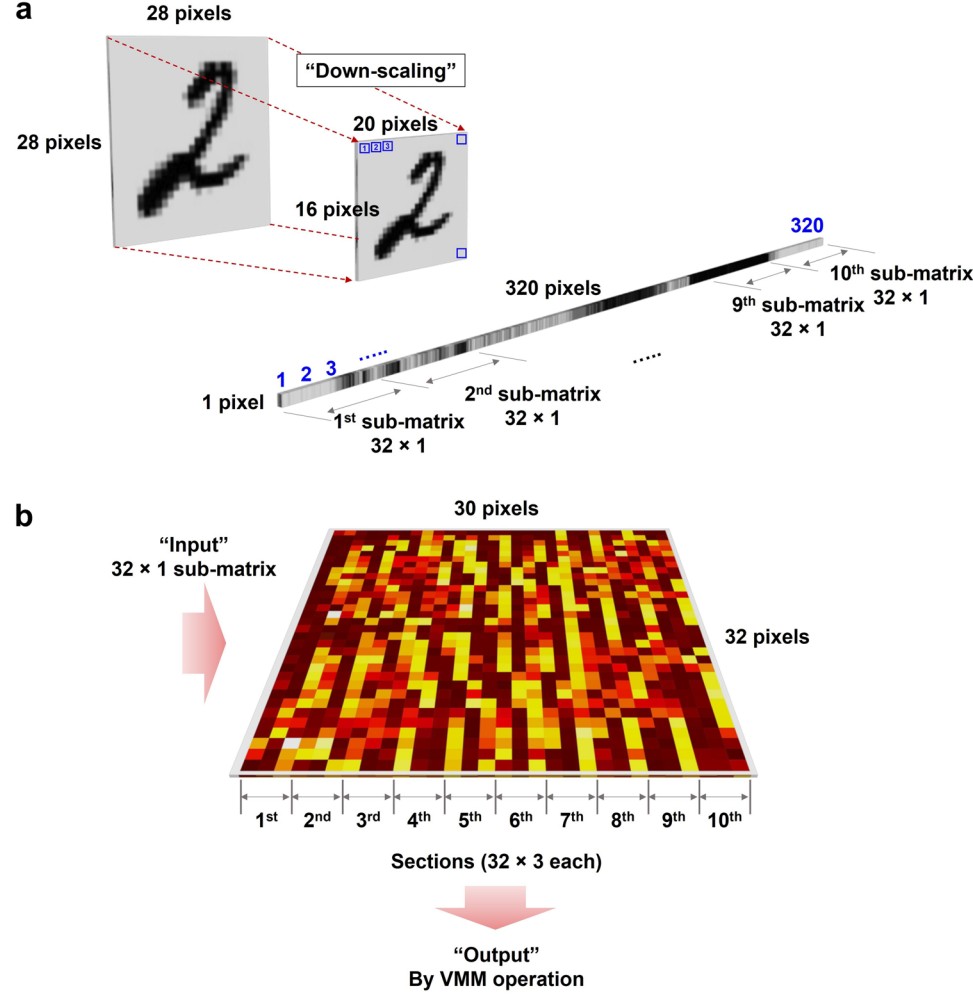

**Fig. 3 | Overall hardware-based classification process.** For hardware-based classification, a 32 × 30 array, representing a partial array of the 1 kb CA, was utilized. To classify the three digits (0, 1, and 2), the 32 × 30 array is partitioned into 10 sets of 32 × 3 matrices. Each column in a submatrix (32 × 1) indicated the three digits 0, 1, and 2. **a** The training and test (inference) datasets were processed by software simulation. By downscaling the MNIST data, originally consisting of 28 × 28 pixels, to 20 × 16 pixels to fit into the 32 × 30 array. To input the images into array for training and inference tasks, the down-sampled images were converted to 320 × 1 matrix and partitioned into 10 sets of 32 × 1 matrices. The training process was performed using the resized data in the software, and the trained data were converted into target conductance (resistance) values to obtain the distribution of trained weights. Based on these weight values, each cell in the 32 × 30 CA was programmed. **b** For the inferencing operations, the resized data were converted into 320 × 1 input vectors with binary values of 0 and 1. These input vectors were then converted into voltages and applied to the memristor matrix in 10 sets. The VMM operation was performed, and the maximum values from the 10 sets were classified as the outputs of the inferencing operations. The digit with the maximum output signal was classified according to the max-current sensing rule[36].

the CA (Fig. 2b). During the electrical measurements, a positive bias sweep from 0 to 3.2 V induced a resistance transition from the high-resistance state (HRS) to the low-resistance state (LRS) (SET, denoted by the black arrow in ①), while a negative bias sweep from 0 to −3.2 V induced the inverse resistance transition (RESET, denoted by the black arrow in ②). All cells in the CA showed a fine distribution without operating failure. To enable the quantitative evaluation of these cells, we investigated the current distributions of the HRSs and LRSs at a reading voltage of 2 V using Eq. 1.

$$\text{Coefficient of variation} = \delta/\sigma \qquad (1)$$

where $\delta$ and $\sigma$ represent the standard deviation and mean reading current, respectively. As shown in Fig. 2c, the reading currents of the HRSs and LRSs were cumulatively plotted, and the coefficient of variations of these states were evaluated as 0.060 and 0.057, respectively. These values indicate that our SRMs in the CA exhibit an operational distribution of less than 6%.

The self-rectifying feature of the memristor, integrated into the crossbar array (CA) with a selectivity of approximately $10^4$ according to

the 1/3 biasing scheme, effectively suppresses sneak currents. Additionally, we observed that the non-filamentary (interface) switching characteristic contributes significantly to minimizing variations between cells. The analyses of the self-rectifying feature and non-filamentary (interface) switching characteristic of the integrated memristor are detailed in our prior research[46]. Moreover, memristors operating in the low current range often exhibit a phenomenon of current drop at specific voltages, and in this memristor-based crossbar array, a significant current drop was observed at −2 V and 1 V. This is not an inherent characteristic of the device but rather a phenomenon stemming from the resolution of the measurement equipment. Although current drops may occur due to the high resolution for low currents, it is important to note that this is not a phenomenon influencing the actual characteristics of the device[48–50].

We tested the electrical pulse (AC)-based operational parameters of our SRM. Figure 2d illustrates the voltage-biasing conditions of each signal line to estimate the parameters of the AC-based operating voltages for the SET and RESET operations in the CA. A train of incremental step pulses proportional to $V_{op}$ was applied to each signal line following a one-third biasing scheme. The same biasing scheme was

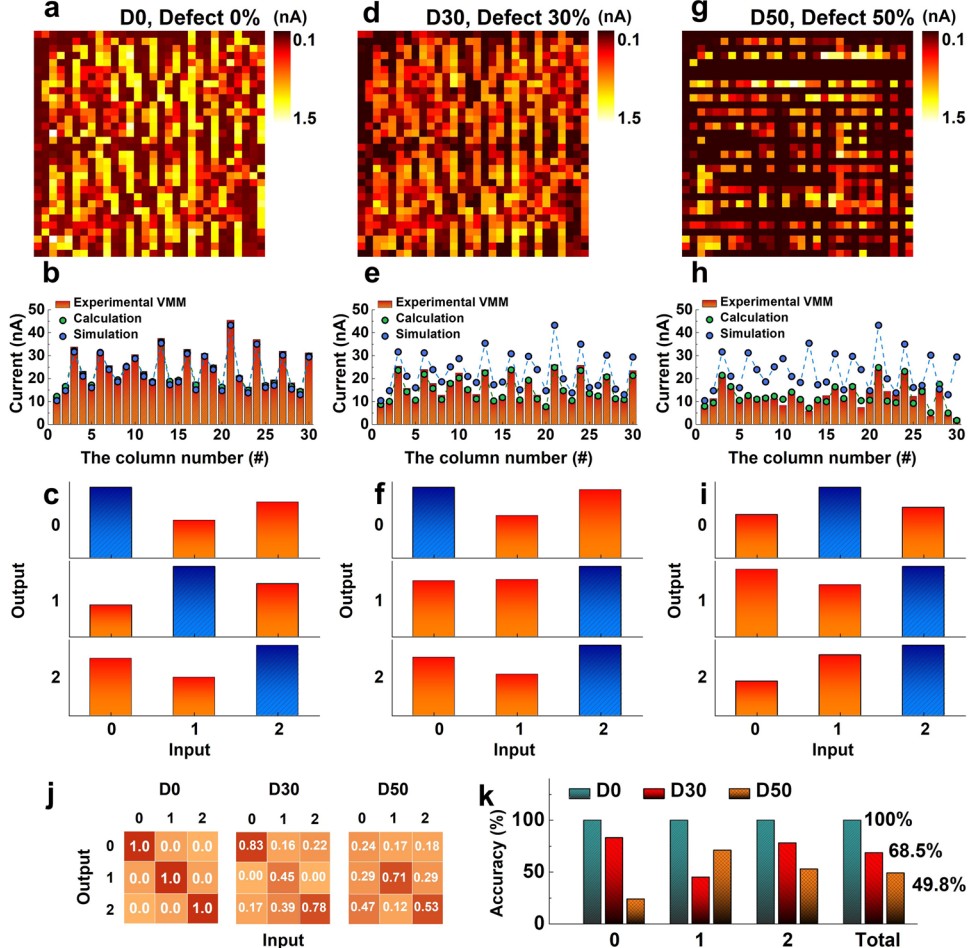

**Fig. 4 | Fully hardware-based demonstration of a single-layer neural network for MNIST data classification and investigation of the impact of the defectiveness of the CA on the classification accuracy. a** Trained weight-mapping results of the D0 CA. After the training process, all cells in the CA were read out at a 2 V reading voltage. **b** VMM operation results of the D0 CA focusing on three parameters: VMM operation results (red bars), trained weight summations (green circles), and simulated weight summations (blue circles). The discrepancy between the VMM operation results and calculated values indicates the feasibility of the VMM operation, while the difference between the VMM operation results and the simulated values indicates the training accuracy. **c** Experimental demonstration of the fully hardware-based classification of the MNIST data using the D0 CA. One of the classification results was showcased by following the max-current sensing rule. The classification result is indicated by the blue bars, which represent the maximum

values of the sensed signals. **d** Trained weight-mapping results, (**e**) VMM results focusing on the three parameters described in (**b**), and (**f**) representative classification results of the D30 CA. While the digits 0 and 2 were classified correctly, the digit 1 was misclassified as 2. **g** Trained weight-mapping results, **h** VMM results focusing on the three parameters described in (**b**), and (**i**) representative classification results of the D50 CA. **j** Classification accuracy for each digit based on 1500 classifications. The classification accuracy of the D0, D30, and D50 CAs for each digit is shown from left to right. **k** Classification accuracies for each digit in different defective CAs and the total classification accuracy. The total accuracies for each defective CA indicated that 100% accuracy was achieved in the D0 CA. However, the accuracies decreased to 68.5% in the D30 CA and 49.8% in the D50 CA. These results demonstrate the practical impact of defects in the CA, particularly open-circuit defects, which significantly degrade the classification accuracy.

applied to all cells in the CA. To verify the resistance state of the SRM, a DC reading voltage was alternately applied after each AC pulse.

The pulse duration of the inhibiting biases was set so that it was sufficiently long to cover that of $V_{op}$, thereby ensuring the proper application of the biasing scheme. If the pulse duration of the inhibiting bias is identical to or shorter than that of $V_{op}$, the simultaneous application of pulses can result in an uncontrollable current overshoot within the cells and the electrical hard breakdown of the cells in the CA. To address this issue, we set the pulse durations of $V_{op}$ and the two inhibiting pulses to 200 and 500 μs, respectively. The AC-based switching characteristics of 30 randomly selected cells were then investigated (Fig. 2e). The measurement results for the SET and RESET operations are shown in the left and right panels, respectively. The SRM exhibited a gradual resistance change during both the SET and RESET operations in accordance with the incremental pulse train. This characteristic is favorable for programming an arbitrary intermediate-resistance state within the designated reading-current margin.

The voltage distributions during each SET and RESET operation are plotted in Fig. 2f. We determined the AC-based SET and RESET voltages for achieving the current values as those obtained from the DC $I–V$ measurements (1.5 and 0.1 nA for SET and RESET, respectively). The mean values of the AC-pulse amplitudes for the SET and RESET operations were 4.3 and −4.9 V, respectively, with an identical standard deviation of 0.1 V. According to Eq. 1, we determined that the coefficient of variations of the operating voltages for SET and RESET were 0.023 and 0.020, respectively (variation, ~2%). These coefficients reflect the fine distribution characteristics of the operating voltages of our SRM, which is favorable for achieving a high CA operating yield.

Considering advancements in electronic devices, a program latency in the range of hundreds of microseconds is generally deemed significant. Despite such notable program latency, the read latency of the SRM was measured at below 10 μs, indicating comparatively lower latency[46]. In the realm of accelerating ANN applications, read operation latency takes precedence over program operation latency, given

the more frequent occurrence of read operations. While the read delay of a single SRM unit in the numbers of microseconds may appear considerable compared to established electronic devices like dynamic RAM, the parallel VMM operation of the array device engaged in ANN computing acceleration can relieve the overall performance degradation from the relative high latency of single SRM.

## Fully hardware-implemented classification with defective CAs

To identify the feasibility of our proposed SRM-based 1 kb CA for ANN applications, we performed a single-layer NN inferencing operation for handwritten-digit classification. Figure 3 illustrates the scheme of the classification task. In this study, we used the MNIST datasets, specifically the digits 0, 1, and 2, for the classification task. Software-based training was performed using a $32 \times 30$ sub-array of the 1 kb CA. To match the dimensions of the CA network, the MNIST images were resized from $28 \times 28$ pixels to $20 \times 16$ pixels. During the training process, the resized MNIST data was reshaped into a $320 \times 1$ matrix. Since the CA density was insufficient to accommodate 320 inputs, we divided the $32 \times 30$ matrix into ten sets of $32 \times 3$ matrices. Therefore, the reshaped $320 \times 1$ images were partitioned into 10 sets of $32 \times 1$ as shown in Fig. 3a. The final weight map representing the conductance of the SRM for image learning was obtained after software-based training. The detailed procedures for acquiring the software-based training set and programming the weight map in the CA are described in the Methods section.

Next, an image classification test was performed using a similar CA-based training method. In brief, input images of $20 \times 16$ pixels (i.e., the test sets) were transformed into a $320 \times 1$ vector, as illustrated in the Fig. 3a. The input vectors were then partitioned into 10 sets of $32 \times 1$ matrices, utilizing binary input values of 0 and 1 to represent black and white regions, respectively. Although the real images in the test sets exhibit analog-like variations in grayscale intensity, the electrical pulses used for the input signals must be reduced to binary signals of 0 and 1 because the number of AC pulse generators available is limited. The input vectors for test sets 0, 1, and 2 were sequentially applied to the trained CA network to perform the VMM operation in each column as illustrated in Fig. 3b. Each column of $32 \times 3$ matrix was indexed to 0, 1, and 2 digits, respectively. The output current values acquired from each column ($32 \times 1$) through the VMM operation were individually summed and compared to determine the maximum current value, which represents the result of the inferencing operation based on the max-current-sensing rule.

Fully hardware-based image classification was performed using the aforementioned methods. To investigate the effect of defective cells in the CA on the accuracy of classification operations, we utilized three CAs with defect portions of 0% (D0), 30% (D30), and 50% (D50). We assumed that the individual defective cells in the CA were in the form of electrically open-circuit cells, and set the conductance of the open-circuit cell to the HRS of each SRM.

Figure 4a, d, g display the weight-map distributions of the trained CA in the cases of D0, D30, and D50, respectively. Upon verifying the weight mapping results, we observed that all cells were accurately trained in the case of D0. However, in the cases of D30 and D50, a corresponding number of defective cells (indicated in black) remained in the HRS. The cells remaining in the HRS represent non-switchable cells in the CA that cause a decrease in CA yield. To evaluate the training accuracy based on the defective cell portion in the CAs, we performed the VMM operation. Figure 4b, e, h display the VMM operation results obtained using CAs with D0, D30, and D50, respectively. Each figure includes three plotted results: the experimental VMM operation result (represented by the red bar), the calculated VMM result based on the programmed weight of each CA column (represented by the green circle), and the calculated VMM result based on the simulated weight values (along the case of D0) obtained through software processing (represented by the blue circle). The

'Experimental results', 'Calculated results', and 'Simulated results' indicate the output current sensed (experimentally measured) from the VMM operations, the sum of read current (conductance) in programmed CA (numerical calculation of current sum at each column based on the programmed (already known) conductance states), and the sum of conductance trained through simulation (the required VMM result to achieve the inference accuracy of 100%), respectively. All these parameters represent values sequentially sensed and calculated for each column. The detail of VMM operation using CA is demonstrated in the 'Method' section. The experimental and calculated VMM results matched well regardless of the defective cell portion, demonstrating that each CA case (D0, D30, and D50) is suitable for the VMM operation without any interference between cells owing to the characteristics of the SRM. However, we found significant differences between the experimental and simulated VMM results in the cases of D30 and D50, which implies that defective cells in the CA can result in its inaccurate training.

Based on the three proposed CA types, fully hardware-based classification operations were conducted using the MNIST dataset. In this task, 1500 classifications were attempted (500 inferences for each digit: 0, 1, and 2) to ensure classification accuracies. Figure 4c, f, i show the representative classification results obtained using CAs of D0, D30, and D50, respectively. We applied the max-current sensing rule mentioned in Fig. 3, and denoted the maximum values of the experimental VMM for the classification tasks using blue bars. The CA of D0 produced output results that were consistent with those of the input (Fig. 4c). However, the CA of D30 showed a misclassification for digit 1, and the CA of D50 showed misclassifications for digits 0 and 1 (Fig. 4f, i), respectively.

The statistical results for classification using CAs of D0, D30, and D50 are summarized in Fig. 4j. The CA of D0 achieved a classification accuracy of 100%, indicating reliable performance. However, the CAs of D30 and D50 showed degradations in classification accuracy for each input digit. Specifically, the classification accuracies of D30 and D50 for the digit 0 were 83% and 24%, respectively. Similarly, the classification accuracies of D30 and D50 for digit 2 were 78% and 53%, respectively. Such findings demonstrate a consistent trend of reduced accuracy in the presence of defective cells. However, a deviation from the monotonous degradation trend of classification accuracy was observed for input digit 1. The classification accuracies of D30 and D50 for digit 1 were 45% and 71%, respectively, indicating that accuracy was not solely dependent on the portion of defective cells in the CAs for specific target classification. The overall classification accuracies for 1500 classification tasks are summarized in Fig. 4k. The CA of D0 achieved a classification accuracy of 100%, whereas the CAs of D30 and D50 achieved classification accuracies of 68.5% and 49.8%, respectively. These results further highlight the impact of defective cells on the classification accuracy of CAs. Furthermore, the impact of defects, specifically those involving LRS stuck cells, was thoroughly investigated, as illustrated in Supplementary Fig. S6.

## Impact of the SRM reading margin on classification accuracy

A large reading margin, which is represented by the resistance ratio between the HRS and LRS, is regarded as an essential property of synaptic memristive devices to achieve high accuracy in classification tasks because it can provide a low error rate to achieve a certain intermediate conductance state within a given conductance range[51]. In this context, conducting filament (CF)-based memristive devices have received considerable interest owing to their high resistance ratio despite their poor operating uniformity. Conversely, non-filament-type (interface-type) memristive devices exhibit highly uniform operating characteristics with a relatively low resistance ratio[52,53]. If the reading margin of a given memristive device is not a crucial factor for achieving high accuracy in classification tasks, interface-type

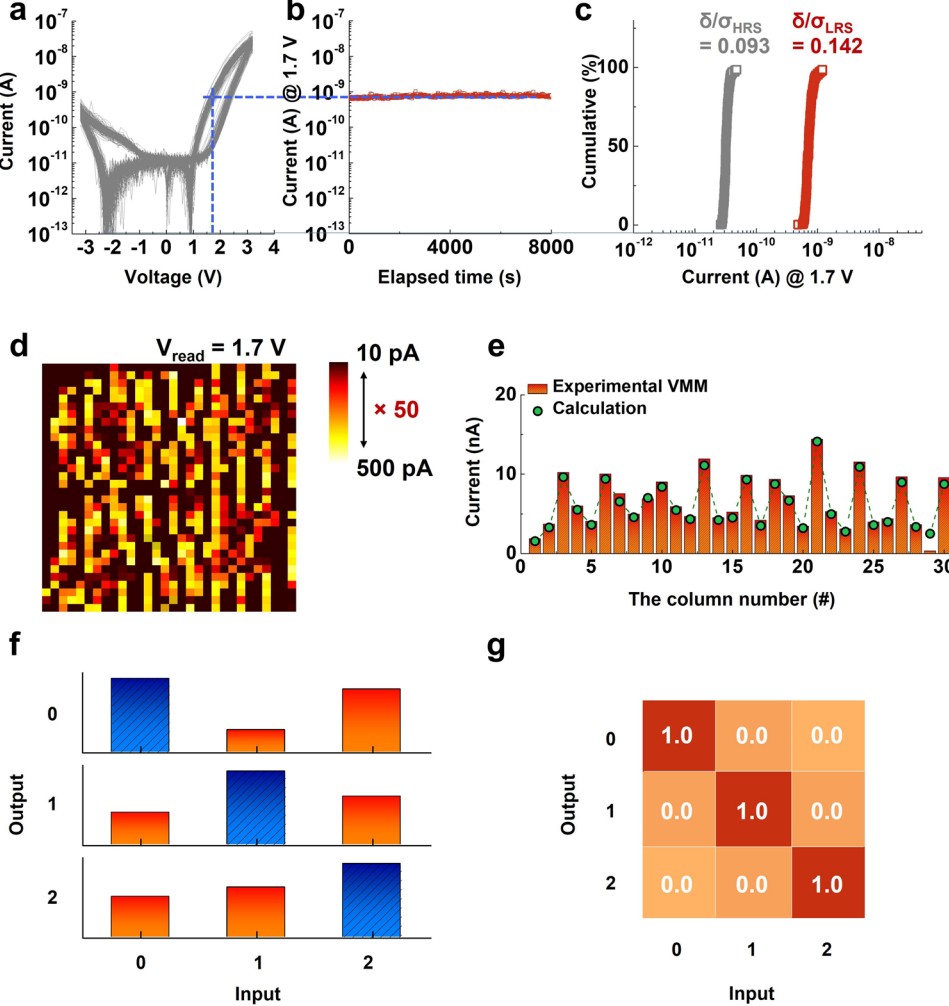

**Fig. 5 | Investigation of the impact of the read margin on classification accuracy. a** DC $I–V$ characteristics of the SRM-integrated CA, and (**b**) retention characteristics measured at a reading voltage of 1.7 V. Robust nonvolatility was observed at 1.7 V. **c** Read current distributions of the LRS and HRS at 1.7 V. The coefficient of variations of the HRS and LRS (0.093 and 0.142, respectively) were relatively low. **d** Trained weight-mapping results obtained at a reading voltage of 1.7 V. **e** Comparison of the VMM operation results and calculated weight summations in each column. The feasibility of VMM operations using a 1.7 V reading voltage was confirmed. **f** Representative classification results of the CA under a 1.7 V reading voltage, and (**g**) total classification accuracy for each digit. All digits were classified accurately (1500 classification operations). Comparison of the two different reading margins (×50 @ 1.7 V and ×15 @ 2.0 V) revealed that the reading margin of the nonfilamentary-type memristor had an insignificant effect on the classification accuracy because of its highly uniform operating characteristics.

memristive devices may be a stronger candidate than CF-based ones for such tasks.

To demonstrate the impact of the reading margin on the classification tasks, we conducted investigations using different reading margins by employing a reading voltage of 1.7 V. Prior to applying a reading voltage of 1.7 V, we verified the retention characteristics of our SRM using the same voltage. Figure 5a, b display the DC $I–V$ curves of the SRM and its corresponding retention characteristics, respectively, as verified under a reading voltage of 1.7 V. The programmed LRS of the SRM was robustly maintained without significant degradation over a 2 h period (initial and final LRS currents of 750 and 742 pA, respectively). The variability among 1024 cells (D2D variability) at a reading voltage of 1.7 V was also confirmed (Fig. 5c). The coefficient of variations (Eq. 1) for the HRS and LRS at 1.7 V were 0.093 and 0.142, respectively, indicating reliable characteristics suitable for VMM operations.

Based on the pretested reliability of our SRM, the CA was trained using a reading margin of 1.7 V (Fig. 5d). During the training process, we set the lowest and highest current values to 10 and 500 pA, respectively, representing an expanded reading margin (resistance

ratio: 50) compared with the case in which a reading voltage of 2.0 V is applied (resistance ratio: 15, Fig. 4a). To confirm the operational reliability of the VMM using the CA, we compared the experimental and calculated VMM results in each column (Fig. 5e). We observed negligible discrepancies between the two, thus confirming the feasibility of the VMM operation at a reading voltage of 1.7 V.

To assess classification ability under different reading margins, we conducted the same tasks (Fig. 4). We performed 1500 classifications (500 inferences for each digit: 0, 1, and 2) to ensure statistical reliability. The representative classification results presented in Fig. 5f demonstrate accurate classification for all digits. Figure 5g shows the statistical outcomes of the 1500 classifications tasks, all of which reveal a classification accuracy of 100%. These experimental findings suggest that the reading margin (conductance range) of a given memristive device has an insignificant effect on the classification task. Thus, interface-type memristive devices may be a strong candidate for ANN applications. In addition, to compare with the smaller read margin condition of CA, the classification results using D30 and D50 for x 50 read margin condition are demonstrated in Supplementary Fig. S7.

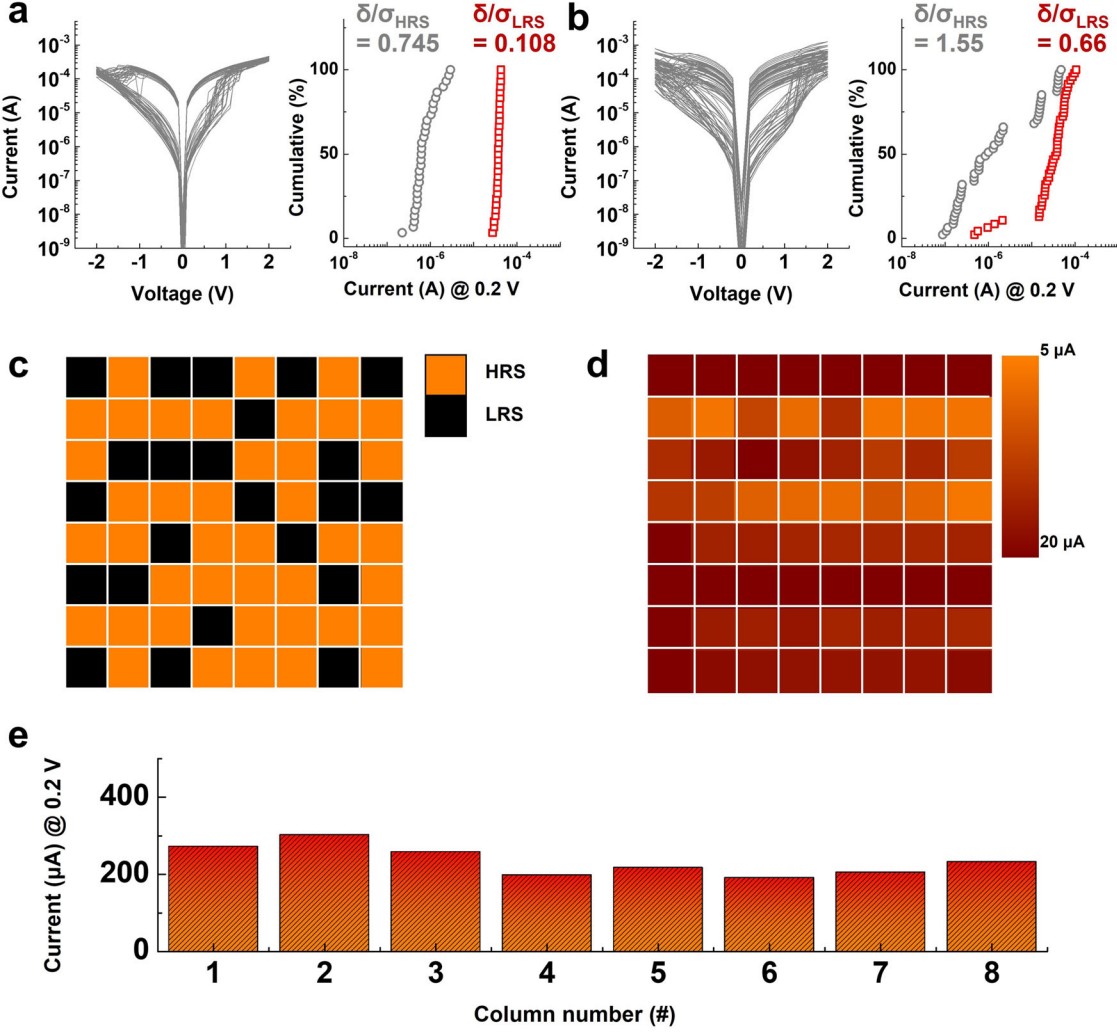

**Fig. 6 | Investigation of the impacts of a non-selective memristor integrated into the passive CA on VMM operation accuracy. a** DC *I–V* characteristics of a unit memristor device without selection functionality (left). Ten randomly chosen cells were measured over three cycles (total number of DC *I–V* curves: 30). The read current distributions of each state were extracted from the DC *I–V* curves (right). **b** DC *I–V* characteristics of an 8 × 8 CA integrated with the memristor. All cells in the CA were measured in one cycle in DC *I–V* sweep mode (graph). The read current distributions of each state were extracted from the DC *I–V* curve (right). **c** Programming scheme of the 8 × 8 CA. A total of 32 randomly chosen cells were programmed to the LRS (black), and the rest of the cells were maintained in the HRS (orange). **d** Resistance distribution of the 8 × 8 CA after the programming step. The programmed weight values of the CA were not coincident with the proposed programming scheme, indicating that programming and reading were not processed accurately owing to sneak current effects. **e** Results of VMM operations using the 8 × 8 CA. Nearly identical output currents were observed in each column, which implies that the VMM operation was not processed accurately because of significant interferences between integrated cells in the CA.

## Impact of selection functionality in a memristive device on classification accuracy

Here, we experimentally demonstrate the significance of selection functionality in memristive devices integrated into the CA to establish an ANN accelerator. For these demonstrations, we prepared a memristive device without selection functionality (symmetric operating current levels in both bias regions) as a control sample. Figure 6a shows the DC *I–V* characteristics of 30 RS cycles (three RS cycles of 10 unit cells each) and the cumulatively plotted reading current variation at 0.2 V. The coefficient of variations (Eq. 1) of the HRS and LRS extracted from the cumulative plot were 0.745 and 0.108, respectively, which are relatively larger values compared with those of the SRM.

Subsequently, the memristive device was integrated into an 8 × 8 CA to examine the electrical characteristics of a CA integrated with a memristive device without selection functionality. During the electrical characterization of the 8 × 8 CA, a one-third biasing scheme, identical to that in Fig. 2a, was applied. As a pretest, two types of bias schemes were examined, and the experimental results are shown in

Supplementary Fig. S3. Figure 6b shows the DC *I–V* characteristics of 64 cells in the 8 × 8 CA and the cumulatively plotted reading current variation at 0.2 V. A significantly higher operational dispersion of the CA was observed in the DC *I–V* characteristics. Furthermore, the coefficient of variations (Eq. 1) of the HRS and LRS were 1.55 and 0.66, respectively; these values are 2.08 and 6.11 times larger than those of the unit device, respectively.

To reveal the feasibility of VMM operation using the 8 × 8 CA with the memristive device, we arbitrarily programmed individual cells in the 8 × 8 CA to the LRS or HRS (Fig. 6c). A total of 23 randomly selected cells were programmed into the LRS using the DC bias-sweep method, and the other cells were maintained in the initial state of the HRS. Subsequently, we verified the resistance state of each cell in the 8 × 8 CA under a reading voltage of 0.2 V and the corresponding one-third biasing scheme. Figure 6d shows the verified resistance state distribution of the 8 × 8 CA, which exhibits a resistance state distribution that is obviously different from that shown in Fig. 6c. While the extracted reading currents of the HRS and LRS from the unit cell in

Fig. 6a are 0.6 and 30 µA, respectively, the observed reading current distributions in the 8 × 8 CA are in range of 5–20 µA. Nearly all cells exhibited a reading current distribution close to that of the LRS, which implies that correct programming or reading operations could not be achieved.

Figure 6e shows the VMM operation results of the 8 × 8 CA. The targeted resistance state distribution obviously differs between each column (Fig. 6c), but the experimental VMM exhibits nearly invariable values in each column, thereby revealing the infeasibility of the VMM operation in the CA without selection functionality. This malfunction in the VMM operation can be observed even at a small integration density of 8 × 8 CA. We also tested a 32 × 32 CA using a memristive device without selection functionality (Supplementary Fig. S4), and observed a large operational distribution in the DC I−V characteristics. Thus, accurate VMM operations for the ANN accelerator cannot be guaranteed when the memristive device integrated into the CA lacks selection functionality.

## Discussion

In this study, we propose a fully hardware-based image classification method using SRM devices integrated into a 1 kb CA. The successful demonstration of image classification tasks using the CA is attributed to the highly reliable characteristics of the integrated SRM. Although general memristive devices are regarded as immature devices for implementation in largely integrated devices owing to their stochastic nature, we successfully achieved reliable characteristics in our memristive device and integrated the SRM into a 1 kb CA without the rendering of a transistor. The most important characteristic of the SRM is its non-filamentary switching operation, which can ensure a narrow operational distribution for high yields in the CA. By excluding the initial electroforming process, we can achieve favorable cycle to cycle (C2C) and D2D characteristics; by contrast, CF-based memristive devices suffer from a large operational distribution. Whereas CF-based memristive devices generally show large reading-margin and robust retention characteristics, non-filamentary-type devices generally exhibit a low reading-margin and poor retention characteristics[48,54,55]. However, the weaknesses of non-filamentary-type memristive devices can be improved by modifying the materials and device structure. In our previous research, we achieved reliable retention characteristics even in the low-operating-current regime (<100 nA) by adopting functional layers (i.e., an oxygen reservoir and oxygen-diffusion barrier layers) in the SRM. In this study, we attempted to demonstrate that another weakness of non-filamentary-type memristive devices, that is, their low reading-margin, is not a crucial factor for image classification tasks. As confirmed by our experiments, variations in reading margin did not influence the inferencing accuracy in image classification tasks.

It has been thought that the CAs are able to tolerate defects in performing VMM operations owing to their severely parallel structure. Thus, even if some defective points are present in the CA, the array can render the defective cells redundant and achieve the correct answer through other signal pathways. However, the experimental demonstration in this study showed that defective cells in the CA had a significant effect on the VMM operation and the accuracy of image classification tasks. Note that we set the defective cells in the CA to its HRS, assuming an open electrical circuit. However, if some defective cells in the CA are in an electrical short circuit, the image classification tasks would show an additional degradation in accuracy. Therefore, for reliable CA operation, the operational yield of individual memristive devices in the CA should be close to 100%. In this case, non-filamentary-type memristive devices may be a good candidate because of their high yield and reliability.

In addition, considering the classification accuracy with terms of bit precision in our system, it is essential to clarify the difference between the nature of the proposed CA conventional digital architectures. While typical digital systems rely on bit precision, the proposed CA operates primarily on analog signals. This fundamental difference means that our approach does not directly engage with bit precision in the traditional sense. However, it is crucial to understand the steps taken to align the MNIST data set with the CA's operational mode.

MNIST's input was reduced to 320 units and normalized pixel values were rounded off to 0 or 1 to simulate the CA's input. Though digital, this process merely digitized input, not weights or CA procedures. The device array utilizes experimental weights adjusted to match its conductance and input current scales. This ensures the CA computes using genuine analog values. To elaborate on the learning process, the reduced input of 320 was chunked into groups of 10 and fed to a three-line array. Each segment has 32 inputs and 3 outputs. Multiplying by weights yields the handwriting classification result for each 10-part segment. The final input is estimated using a soft-max function after aggregating these scores. The calculated weights were converted to the target conductance and applied to the CA using the following formula:

$$G_T = W \times (G_{LRS} - - G_{HRS}) \qquad (2)$$

Let, the conductance for training = $G_T$, simulated synaptic weight = W, $G_{LRS}$ = LRS conductance of CA, and $G_{HRS}$ = HRS conductance of CA.

Regarding the weights obtained through software, they are represented using the float 32 data type, offering sufficient precision for our needs. Given that the device responsible for applying conductance to the array can handle a range from 0.1 A to as precise as 0.1 pA, there should be confidence in the CA's conductance resolution to represent and manipulate these synaptic weights accurately. Furthermore, in terms of the required precision for NN applications, it varies based on the specific task and network architecture. However, with our approach and the inherent precision of the CA's design, we believe that the system is well-equipped to meet and potentially exceed the precision requirements commonly associated with NN applications.

The memristive device in the CA must have selection functionality to ensure reliable VMM operations based on the CA. This functionality is necessary for both training and inferencing, as it enables correct and accurate operations using the CA. In this study, we comparatively investigated the operation of a CA integrated with a memristive device without selection functionality. Although our intention was to obtain randomly distributed resistance states for the HRS and LRS in the 8 × 8 CA (Fig. 6c), the verification results (Fig. 6d) differed significantly from the initially intended resistance state distribution. Previous research often employed 1T1M-structured unit cells to implement selection functionality in each cell in the CA. Given the increasing demand for a vast memory capacity in future computing technologies to process large amounts of data, a singular form of selection and memory can provide an advantage in terms of the dense integration of electronic devices. From this perspective, a non-filamentary switching-type SRM with high selectivity offers reliable performance and robust reproducibility.

In summary, we proposed a 1 kb CA integrated with SRMs as a hardware accelerator for ANN applications. Image classification tasks using our hardware accelerator revealed a remarkable recognition accuracy of 100%, which is comparable with that of software-based simulations. Furthermore, the impact of non-ideal factors on the hardware accelerator was investigated. We highlighted the merits of hardware accelerators based on SRMs. Our accelerator demonstrated uniform operational characteristics across all devices in the CA, with extremely low defectiveness (resulting in high yields) during fabrication and high immunity to sneak currents from neighboring cells. We believe that non-filamentary-type-based SRMs can serve as a key device component in future ANN applications. We also provide

**Table 1 | The comparison results of hardware implementation for neural network applications using memristive crossbar arrays**

|  | This work | Ref. 57. | Ref. 36. | Ref. 30. | Ref. 31. | Ref. 56. |
|---|---|---|---|---|---|---|
| Array configuration | 1 R (SRC) | 1S-1R | 1T-1R | 1T-1R | 1T-1R | 1T-1R |
| Array density | 32 × 32 (1 kb) | 9 × 9 (<1 kb) | 128 × 64 (8 kb) | 54 × 108 (~5 kb) | 128 × 16 (2 kb) | 128 × 64 (8 kb) |
| Array yield (%) | 100 | 100 | 98.9 | Not available | 99 | Not available |
| Energy efficiency (TOPS/W) | 4.35 | Not available | 115 | 1.37 | 11 | 77.4 |
| Application | MNIST classification | Pattern classification | MNIST classification | Pattern classification | MNIST classification | MNIST classification |
| Accuracy | 100 | 100 | 89.9 | 94.6 | 96.19 | 92.3 |
| Integrated device stack | Ru/HfSiO$_y$/Al$_2$O$_3$/HfSiO$_x$/TiN | Pt/Ru/TiO$_2$/RuO$_2$/Pt/HfO$_2$/TiN | Ta/HfO$_x$/Pd | Pd/WO$_x$/Au | TiN/TaO$_x$/HfO$_x$/TiN | Pt/TaO$_x$/Ta |

valuable insights for addressing reliability issues associated with memristive devices.

Through this study, the NN applications utilizing memristive crossbar arrays are compared in Table 1. It is noteworthy that our study represents the first reported demonstration in the field of SRMs, utilizing a purely self-rectifying memristor-based passive crossbar array for hardware implementations of NN applications. While the energy efficiency of the proposed CA is currently lower than ref. 36,56., we anticipate that higher efficiency can be achieved by increasing the array density.

Furthermore, we compare the energy efficiency of our hardware accelerator with traditional microelectronic chips such as GPUs, FPGAs, and ASICs in Table 2. The results of the comparison show that our hardware accelerator achieves relatively high energy efficiency due to significantly lower power consumption compared to traditional microelectronic chips. For detailed calculations, refer to Supplementary Note 1.

## Methods

### Device fabrication
A SiO$_2$/Si substrate was coated with a 200-nm-thick TiN layer by sputtering. The TiN layer was then patterned onto the CA bottom electrode (BE) using conventional photolithography and inductively coupled plasma (ICP)-reactive ion etching. The ICP and substrate bias powers were maintained at 200 and 20 W, respectively, during the TiN etching process. Etching was performed using Ar and Cl$_2$ gases at flow rates of 5 and 30 standard cubic centimeters per minute, respectively. The process temperature was kept at 25 °C using a water-circulation cooling system, and the etching rate was approximately 70 nm/min. Photoresist (PR) residues remaining on the patterned TiN BE were removed using acetone, followed by cleaning with isopropyl alcohol and deionized water. Next, 1-nm-thick Hf$_{0.5}$Si$_{0.5}$O$_2$ and 2-nm-thick Hf$_{0.8}$Si$_{0.2}$O$_2$ layers were deposited onto the substrate using atomic layer deposition (ALD) with a traveling wave-type ALD system. Deposition was carried out at 250 °C using tetrakis(ethylmethylamido hafnium) and bis(diethylamino)silane precursors as well as H$_2$O and O$_2$ plasma as sources of Hf and Si oxidants, respectively. The ALD process involved supercycle deposition with alternating HfO$_2$ and SiO$_2$ layers. The SiO$_2$:HfO$_2$ cycle ratios of the Hf$_{0.5}$Si$_{0.5}$O$_2$ and Hf$_{0.8}$Si$_{0.2}$O$_2$ layers were set to 1:3 and 1:1, respectively. An intermediate Al$_2$O$_3$ thin-film layer was deposited between the Hf$_{0.5}$Si$_{0.5}$O$_2$ and Hf$_{0.8}$Si$_{0.2}$O$_2$ layers using ALD at 150 °C with trimethyl aluminum and H$_2$O as the Al source and oxidant, respectively. Subsequently, a crossbar-type top electrode (TE) pattern was defined over these layers using photolithography, and a 100-nm-thick Ru layer was deposited on it using DC magnetron sputtering. Finally, a Ru/Hf$_{0.8}$Si$_{0.2}$O$_2$/Al$_2$O$_3$/Hf$_{0.5}$Si$_{0.5}$O$_2$/TiN stacked CA device was fabricated using the conventional lift-off process. After its fabrication, images of the CA were acquired using high-resolution scanning electron microscopy (HR-SEM, S-4800, Hitachi, 3.0 kV, 7 μA)

A TiN BE was formed via the same HfSiO$_x$-based SRM fabrication method to prepare a memristor without selection functionality. Subsequently, a 4-nm-thick GeTe thin film was deposited on the electrode as the active layer using a radio frequency (RF) magnetron sputtering system. A TE pattern was formed using photolithography, and a 200-nm-thick Cu layer was deposited over these layers using DC magnetron sputtering. After the final liftoff process, a Cu/GeTe/TiN-stacked memristor device was successfully fabricated.

### Electrical measurements
A customized CA switching zig, CA probe station, and CA probe card were used to access individual cells in the CA and characterize the CA devices. A semiconductor parameter analyzer (SPA, HP 4155 B) was used as an electrical bias source for the CA. Four channels of bias sources were used to induce the operational bias and ground into the selected lines and the inhibiting biases into the unselected lines, allowing the CA to operate properly. Electrical pulse-based switching characteristics were measured using arbitrary function generators (Agilent 81150 A, Tektronix AFG31102) and RF electric circuit switch boxes. The DC I–V characteristics of the CA were measured in DC I–V sweep mode using the SPA and customized CA characterization equipment. During the electrical measurements, the Ru (or Cu) TE was biased, whereas the TiN BE was electrically grounded. Specific word and bit lines were biased to V$_{op}$ and grounded to access the selected cell in the CA, and the rest of the lines were biased to the inhibiting voltages according to the bias scheme. In this study, 1/3 V$_{op}$ and 2/3 V$_{op}$ were applied as inhibiting voltages.

### Classification of MNIST handwritten digits
**Data preprocessing.** The original 28 × 28 pixel MNIST images were subjected to a center-cropping process to reduce their size to 24 × 24 pixels. This cropping technique helped highlight the central regions of interest. Subsequently, the cropped images were resized to 20 × 16 pixels. This resizing step helped maintain the inherent visual characteristics of the original MNIST images while efficiently utilizing the available data. A pixel-value normalization step was performed to ensure consistent pixel values across the dataset. This step involved rounding the pixel values to either 0 or 1, effectively binarizing the images.

**Software-based data processing and learning of CA.** To emulate the CA's input, the MNIST dataset was modified where its input was reduced to 320 units and normalized pixel values were rounded off to either 0 or 1. This process, though digital in nature, only digitized the input, not the weights or the internal operations of the CA. The device array utilizes experimentally derived weights, which are adjusted to match the conductance scale and input current scale of the array. This ensures that the CA processes genuine analog values throughout its computation. To elaborate on the learning process in more detail: the

**Table 2 | The comparison results with traditional microelectronic chips, which function as hardware accelerators for computation**

|  | This work | Ref. 58. | Ref. 59. | Ref. 60. | Ref. 61. | Ref. 62. | Ref. 63. |
|---|---|---|---|---|---|---|---|
| Hardware | Memristive passive crossbar array | GPU (Titan X) | FPGA (Virtex 7) | FPGA (Arria-10) | FPGA (Zynq-7000) | ASICs (YodaNN) | ASICs (Google TPU) |
| Energy efficiency | 4.35 TOPS/W | 98 GOPS/W | 935 GOPS/W | 30 GOPS/W | 14.3 GOPS/W | 8.6 TOPS/W | 2.3 TOPS/W |

reduced input of 320 was chunked into groups of 10, which were then used as inputs for an array with three lines. As a result, each section received 32 inputs and produced 3 outputs. The output generated by multiplying with the weights represents the score for the handwriting classification within each divided section of 10. These scores are then aggregated, and a soft-max function is applied to estimate the final input.

In the training (learning) and testing (inference) tasks in the software-based classification process, we utilized the MNIST dataset comprising 18623 and 3147 samples for training and testing our single-layer NN, respectively. Subsequently, in the fully hardware-based classification process, we evaluated the classification performance of our developed memristor CA under varying defect ratios and read margins. This assessment utilized a total of 1500 test data points, with 500 data points allocated to each digit.

**Hardware implementation for MNIST-digit classification.** The single-layer NN was trained using preprocessed MNIST digits with a software-based training method. The weight values of the trained network were derived by mapping analog values between binary 0 and 1. These resulting weights for each cell in the Crossbar Array (CA) were analog values ranging between 0 and 1. Subsequently, these weights were converted to corresponding conductance values within the reading current range of the SRM, specifically between 0.1 and 1.5 nA. In this conversion, a value of 0 from the software-based procedure was mapped to a High-Resistance State (HRS), while a value of 1 was mapped to a Low-Resistance State (LRS). Intermediate values between 0 and 1 were proportionally converted to intermediate conductance states between HRS and LRS following Eq. (2) in the 'Discussion' section.

Our digit classification procedure, designed for the CA array, involves dividing the input data into ten distinctive chunks, given our input size of 320. This segmentation is crucial for our architecture, which relies on processing individual data chunks through dedicated modules to leverage the CA array's intrinsic parallel processing capabilities. Each module is assigned the task of capturing unique features from its respective segment of the data. After individual processing, the outputs from all these modules are aggregated, resulting in a combined prediction. This combined prediction is then transformed to provide probability scores for the digits 0, 1, and 2.

Our network architecture is uniquely designed for this data segmentation. Instead of a traditional fully connected approach, we separate the input data into ten smaller chunks, each comprising 32 units. Each of these chunks is processed through a specific module, consisting of a linear layer with 32 input features and 3 output features. The collective output from these ten modules produces the final prediction.

## Data availability
The data that support the findings of this study are available from the corresponding author upon reasonable request.

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

## Acknowledgements

G.H.K. acknowledges support from researches grant from the National Research Foundation of Korea (grant no. 2022M3F3A2A01044952 and 2022M3H4A1A04068923). H.J. was supported by Copyright Technology R&D Program by Ministry of Culture, Sports and Tourism and Korea Creative Content Agency (CR202104002) and the Future Resource Research Program of the Korea Institute of Science and Technology (2E32242).

## Author contributions

K.J. and J.J.R. conducted the device fabrication and tested the electrical characteristics. S.I. and H.J. provided the software-based training and testing sets of MNIST-based hand-written numbers. H.K.S. and T.E. helped the functional thin film deposition processes. M.K.Y., D.S.J., and G.H.K. managed the research project and wrote the paper.

## Competing interests

The authors declare no competing interests.
