## [Peer Review File · Nature Communications]

Purely Self-rectifying Memristor-based Passive Crossbar Array for Artificial Neural Network AcceleratorsREVIEWER COMMENTS

Reviewer #1 (Remarks to the Author):

The manuscript presents a study on a physical memristor-based crossbar array capable of high-accuracy pattern recognition. The advantages of the self-rectifying behaviour, impact of defective cells, reading margin are also investigated. The work is timely and novel, and will be beneficial to the society. The paper is well-written, demonstrating high-quality results and insightful discussions. However, several critical points need to be addressed before considering publication in Nature Communications:

1. The authors should clarify the dimensions of each device (cross-point) and elaborate on their scalability.
2. In Figure 2b, it would be beneficial for the authors to provide a brief commentary on the observed self-rectifying properties in the memristor, along with an explanation for the significant current drop at approximately 1 V and -2 V. Additionally, discussing the basic switching mechanism (interfacial switching) would provide a more comprehensive understanding to the readers.
3. In Figure 2d, since the pulse duration falls within the range of hundreds of microseconds, it is crucial to address the limitations of the pulse duration on this memristor.
4. For Figure 2e, linearity and symmetry are pivotal properties for memristor Crossbar Arrays (CAs) to perform Vector-Matrix Multiplication (VMM). The authors should investigate the gradual resistance change under identical pulse trains and comment on the linearity and symmetry regarding CA performance.
5. The training of the network was executed using software. The authors should offer more information about the training process, such as the number of current states utilized. Furthermore, details about how the trained weights were implemented in the CA should be provided.
6. In Figure 4, it would be insightful to understand if there are any differences between the devices stuck at High Resistance State (HRS) and Low Resistance State (LRS) for the defective devices.
7. For Figure 5, considering that both conditions at a reading margin of 50 yielded 100% accuracy at D0 configuration, it would be more conclusive to compare the performance at D30 or D50 configurations. Alternatively, investigating performance at a lower (<15) reading margin would also provide valuable insights.
8. The selection functionality is pivotal to a selector-free CA. It is essential to elaborate if there are any specific requirements regarding the level of selectivity the self-rectifying property needs to provide for decent performance.

9. In addition to the reading margin and selection, exploring the impact of Cell-to-Cell (C2C) and Device-to-Device (D2D) variations on CA performance is crucial, as both are fundamental characteristics for memristor devices.

Reviewer #2 (Remarks to the Author):

This paper presents a Crossbar Array (CA) based on self-rectifying memristors as hardware accelerators for artificial neural network (ANN) applications. A 1 kb CA is implemented as a hardware accelerator for neural network operations, and the impact of various CA characteristics on image classification tasks is investigated, achieving 100% accuracy in a three-class classification on the MNIST dataset. However, there are some concerns that need to be addressed to enhance the paper's value and clarity.

1. This paper could benefit from a comparative analysis with other similar research in the field. Including a table or graph comparing the performance of the proposed CA with other hardware accelerators for ANNs would help readers understand the significance and value of this work. Such a comparison should be highlighted in the abstract and conclusion sections to emphasize the paper's contributions.

2. This paper claims that the memristor-based passive crossbar array can accelerate neural network operations, but it lacks specific chip performance metrics, such as computational speed and power consumption. Providing these details and comparing the CA's performance with traditional microelectronic chips like CPUs, GPUs, and ASICs would demonstrate its advantages more clearly.

3. This paper mentions additional power consumption due to the application of different voltages to suppress sneak currents. It would be helpful to specify the proportion of this additional power consumption relative to the total power consumption and its impact on the overall chip performance. This information would enable readers to better evaluate the trade-offs involved in using the proposed CA for ANN applications.

4. This paper should provide clearer definitions of the terms "Experimental result," "Calculated result," and "Simulated result." Additionally, quantitative descriptions of how well the experimental and calculated results match, in terms of bit precision, would help readers better understand the accuracy of the proposed CA. The paper should also explain the required bit precision for neural network applications and demonstrate that the CA can meet these requirements.

5. The authors should provide a more detailed description of the digit classification process using the CA array. This should include clarifying whether the network is fully connected or not, and explaining the significance of the localized "winner-takes-all" rule. Additionally, detailed information on the selection rules for the training and testing sets leading to the claimed 100% accuracy in classifying handwritten digits 0, 1, and 2 should be provided. Furthermore, the paper should explain how the

multiplication of currents read from each column of the CA is achieved, including whether digital signals are used, and how this process affects overall performance.

Responses to the reviewers' comments

The manuscript number: **NCOMMS-23-41697-T**

Reviewer #1 (Remarks to the Author):

The manuscript presents a study on a physical memristor-based crossbar array capable of high-accuracy pattern recognition. The advantages of the self-rectifying behaviour, impact of defective cells, reading margin are also investigated. The work is timely and novel, and will be beneficial to the society. The paper is well-written, demonstrating high-quality results and insightful discussions. However, several critical points need to be addressed before considering publication in Nature Communications:

Answer: We thank the reviewer for the positive evaluation. We did our best to address the reviewer's feedbacks.

Q1. The authors should clarify the dimensions of each device (cross-point) and elaborate on their scalability.

Answer:

We appreciate the reviewer's insightful feedback. The authors entirely concur with the reviewer's statement, and we apologize for not mentioning the effective dimension of each device in CA. In this study, the effective line (metal lines of CA) width was 2 μm ; consequently, the effective area of the cross-point was 4 μm^2 . Due to the technological limitations of our laboratory equipment, the photolithography procedure was carried out using UV light, which could hardly reduce the line width to below 2 μm . Consequently, the minimum effective dimension of our device is set to 4 μm^2 .

Author action:

We added the information on the effective dimension of each device in CA into the 'Electrical Characteristics of the SRM-based 1 kb Passvie CA' section on page 6 in the revised manuscript. The added sentence is as follows.

"The line width for each line comprising the top and bottom electrodes is set at 2 μm , resulting in an effective junction area of 4 μm^2 ."

Q2. In Figure 2b, it would be beneficial for the authors to provide a brief commentary on the observed self-rectifying properties in the memristor, along with an explanation for the significant current drop at approximately 1 V and -2 V. Additionally, discussing the basic switching mechanism (interfacial switching) would provide a more comprehensive understanding to the readers.

Answer:

We thank for the helpful comments provided by the reviewer. In response to the reviewer's feedback, we would like to clarify the characteristics of the self-rectifying memristor integrated into CA. Under a 1/3 biasing scheme, the memristor exhibited a relatively high selectivity, as demonstrated in Fig R1a, with a selectivity value of approximately 10^4 .

Furthermore, the rectifying characteristic of the memristor can be achieved by using electrodes with asymmetric work functions to form an asymmetric Schottky barrier as demonstrated in our previous research⁴⁶. The resistance change occurs due to the variation in the Schottky barrier height, which is induced by the applied voltage and oxygen ion movement, as determined through fitting based on the Schottky emission conducting mechanism as depicted in Fig R1b. This fitting result revealed the presence of rectification due to the asymmetric electronic barrier. Furthermore, the self-rectifying memristor exhibits interface switching characteristics, as evidenced by area-dependent measurements. As illustrated in Fig R1c, the device's current increase with increasing area confirms the typical behavior of an interface switching-type memristor. (It is not filamentary characteristic which shows area-independent variation of operation current because of the localized current paths.)

Figure R1. **a** DC I - V characteristic of the unit memristor and selectivity under the $1/3$ biasing scheme. **b** Conduction mechanism fitting results of the memristor device. **c** Area dependency

of the memristor device

Moreover, the pronounced current drops at -2 and 1 V stem from the characteristics of the measurement equipment. The proposed memristor operates within the low current range, with an 'off leakage current' formed at a level below picoamperes. In previous studies⁴⁶, the DC I - V characteristics of unit devices were evaluated using the semiconductor parameter analyzer (SPA) HP4145. In this work, the SPA utilized for assessing the electrical properties of the crossbar array component is the HP4155, representing an improvement in equipment resolution compared to previous evaluations. Consequently, when measured using HP4145, values for currents below picoamperes appeared as results resembling noise. However, measurements with HP4155, characterized by higher resolution, reveal phenomena such as current drops. Various research results, as evidenced in the references provided, show that memristors operating in the low current range exhibit current drops at specific voltages as shown in ref. R1, R2, and R3. Although current drops may occur due to the low resolution for low currents, this is not a phenomenon influencing the actual characteristics of the device. Again, thank you for the helpful comments.

Ref. R1

Kim, K. M. et al. *Nano Lett.* **16**, 6724–6732 (2016).

Ref. R2

Huang, J. -N. et al. *Solid State Ionics.* **369**, 115732 (2021).

Ref. R3

Choi, S. et al. *NPG Asia Mater* **10**, 1097–1106 (2018)

Author action: Following the reviewer's comments, we added brief commentary about the self-rectifying characteristic and switching mechanism in the revised manuscript on Page 8 in the revise manuscript. The revised main texts are shown below.

“The self-rectifying feature of the memristor, integrated into the crossbar array (CA) with a selectivity of approximately 10^4 according to the 1/3 biasing scheme, effectively suppresses sneak currents. Additionally, we observed that the non-filamentary (interface) switching characteristic contributes significantly to minimizing variations between cells. The analyses of the self-rectifying feature and non-filamentary (interface) switching characteristic of the integrated memristor are detailed in our prior research⁴⁶. Moreover, memristors operating in the low current range often exhibit a phenomenon of current drop at specific voltages, and in this memristor-based crossbar array, a significant current drop was observed at -2 V and 1 V. This is not an inherent characteristic of the device but rather a phenomenon stemming from the resolution of the measurement equipment. Although current drops may occur due to the high resolution for low currents, it's important to note that this is not a phenomenon influencing the actual characteristics of the device^{51, 62, 63}.”

The related studies are added in the ‘Reference’ section in the revised manuscript as below.

62. Huang, J. -N., Huang, H. -M., Xiao, Y., Wang, T and Guo, X. Memristive devices based on Cu-doped NbO_x Films with large self-rectifying ratio. *Solid State Ionics*. **369**, 115732 (2021).
63. Choi, S., Jang, S., Moon, J. H. *et al.* A self-rectifying TaO_y/nanoporous TaO_x memristor synaptic array for learning and energy-efficient neuromorphic systems. *NPG Asia Mater* **10**, 1097–1106 (2018).

Q3. In Figure 2d, since the pulse duration falls within the range of hundreds of microseconds, it is crucial to address the limitations of the pulse duration on this memristor.

Answer:

We thank for the reviewer's critical comment. The proposed SRM cell functions with a total current of less than 100 nA. This cell exhibits a resistance range ranging from hundreds of M Ω to several G Ω , resulting in a significant RC delay. As a result, it demonstrates a program delay of 100 μ s⁴⁶. In cases where the frequency of read operations exceeds that of program operations in accelerating of ANN applications, the latency of the SRM's read operations is more significant factor. It has been measured that the read latency of this component is lower, at 10 μ s, compared to the program latency⁴⁶. Although the read delay of a unit SRM at the level of tens of μ s could seem substantial compared with matured electronic device like dynamic RAM, the degrading of the overall performance of the array device operating acceleration of ANN computing can be relieved because of the parallel nature of the VMM operation. Despite the relatively higher operation latency, the substantially low cycle to cycle (C2C) and device to device (D2D) variation and high selectivity of the SRM can reduce the contribution of latency-induced performance degradation in high-density arrays. Nevertheless, it is acknowledged that microsecond-level latency presents a limitation when replacing mature electronic devices, and our research team is aware of this issue. Hence, we are conducting research with the aim of reducing the operation latency of interface-switching memristors in next scope. Again, we appreciate the reviewer's valuable comment

Author action:

Considering the reviewer's comment, we added the statement below, on Page 9 in the revised manuscript

“Considering advancements in electronic devices, a program latency in the range of hundreds of microseconds is generally deemed significant. Despite such notable program latency, the read latency of the SRM was measured at below 10 μ s, indicating comparatively lower latency⁴⁶. In the realm of accelerating ANN applications, read operation latency takes precedence over program operation latency, given the more frequent occurrence of read operations. While the read delay of a single SRM unit in the numbers of microseconds may appear considerable compared to established electronic devices like dynamic RAM, the parallel VMM operation of the array device engaged in ANN computing acceleration can relieve the overall performance degradation from the relative high latency of single SRM.”

Q4. For Figure 2e, linearity and symmetry are pivotal properties for memristor Crossbar Arrays (CAs) to perform Vector-Matrix Multiplication (VMM). The authors should investigate the gradual resistance change under identical pulse trains and comment on the linearity and symmetry regarding CA performance.

Answer:

We appreciate the insightful comments from the reviewer. As highlighted by the reviewer, the linear and symmetrical conductance changes under identical pulse train conditions are crucial characteristics for VMM operations. In response, the authors have presented measurements of the 'potentiation' and 'depression' characteristics using our developed SRM-integrated CA. The results are depicted in Fig. R2.

To conduct potentiation and depression under identical pulse train conditions, we employed the 'CSPP (constant step pulse program) method. The electrical pulse conditions were set to 200 μs width, 3.7 V for potentiation (SET) amplitude, and -3.5 V for depression (RESET) amplitude. Additionally, 64-level pulse numbers for each potentiation and depression process were induced, representing an optimized number of pulses that can exhibit a nearly ideal symmetric factor for the process.

Furthermore, addressing the reviewer's suggestion, the authors calculated the 'linearity factor' and 'symmetric factor' of the potentiation and depression curves. The calculations were performed using equations (1) and (2), respectively. The results indicate linearity factors of 0.0514 and 0.195 for potentiation and depression, respectively. Additionally, the symmetric factor for these processes is calculated to be 1.15.

Pulse conditions			Linearity factor		Symmetry factor
Width	Potentiation	Depression	Potentiation	Depression	
200 μ s	3.7 V	-3.5 V	0.0514	0.195	1.15

Figure R2. The potentiation and depression characteristics of the proposed CA

Author action:

We add the potentiation and depression measurement results into Supplementary Fig. S5 in the revised ‘Supplementary Information’.

Pulse conditions			Linearity factor		Symmetry factor
Width	Potentiation	Depression	Potentiation	Depression	
200 μ s	3.7 V	-3.5 V	0.0514	0.195	1.15

Supplementary Fig. S5: Synaptic properties measured utilizing the SRM-based 1 kb passive CA.

To investigate the gradual resistance change under identical pulse trains, we conducted the ‘potentiation’ and ‘depression’ characteristics which are the synaptic plasticity utilizing the constant step pulse program (CSPP) method. The electrical pulse conditions induced are set to 200 us of width, 3.7 V of potentiation (SET) amplitude, and -3.5 V of depression (RESET) amplitude. Furthermore, the 64-level pulse number for each potentiation and depression process is induced, which is the optimized number of pulses that can exhibit a nearly ideal symmetric factor for the process. To calculate the linearity factor, we utilized the equation (1)

$$G = \lambda \exp(\alpha x + \beta) + \gamma \quad (1)$$

Where G is the conductance, x is the input pulse number, α is a nonlinearity factor, λ , β , and γ are other related coefficients to determine the fitting. [2]

And, the equation (2) is utilized to calculate the symmetric factor of the operation.

$$\text{Symmetric factor} = \quad (2)$$

$$\max \left(\frac{\Delta G_D}{\Delta G_P}, \frac{\Delta G_P}{\Delta G_D} \right)$$

Where ΔG_P and ΔG_D are the overall conductance changes in the potentiation and depression processes, respectively, and the max function is used. [3]

As a result, the linearity factors of 0.0514 and 0.195 for potentiation and depression, respectively, are calculated. Furthermore, the symmetric factor of these processes is calculated for 1.15.

In addition, we added the references related to the used calculations for linearity factor and symmetric factor in the revised 'Supplementary Information' as follow.

2. Chen, J. et al. LiSiO_x-Based Analog Memristive Synapse for Neuromorphic Computing. IEEE Elec. Dev. Letters. **40**. 542-545.(2019)
3. Ryu, J. J et al. Highly Linear and Symmetric Weight Modification in HfO₂-Based Memristive Devices for High precision Weight Entries. Adv. Electron. Mater. **6**. 2000434. (2020).

Q5. The training of the network was executed using software. The authors should offer more information about the training process, such as the number of current states utilized. Furthermore, details about how the trained weights were implemented in the CA should be provided.

Answer:

Thank you for the insightful comment. It's essential to clarify the difference between the nature of the proposed CA and conventional digital architectures when discussing precision. While typical digital systems rely on bit precision, the proposed CA operates primarily on analog signals. This fundamental difference means that our approach does not directly engage with bit precision in the traditional sense. However, it's crucial to understand the steps taken to align the MNIST data set with the CA's operational mode.

Figure R3. Schematic of the software-based training and inference processes

To emulate the CA's input, the MNIST dataset was modified where its input was reduced to 320 units and normalized pixel values were rounded off to either 0 or 1. This process,

though digital in nature, only digitized the input, not the weights or the internal operations of the CA. The device array utilizes experimentally derived weights, which were adjusted to match the conductance scale and input current scale of the array. This ensures that the CA processes genuine analog values throughout its computation.

To elaborate on the learning process in more detail: the reduced input of 320 was chunked into groups of 10, which were then used as inputs for an array with three lines. As a result, each section received 32 inputs and produced 3 outputs. The output generated by multiplying with the weights represents the score for the handwriting classification within each divided section of 10. These scores are then aggregated, and a soft-max function is applied to estimate the final input.

The calculated weights were converted to the target conductance and applied to the CA using the following formula:

$$G_T = W \times (G_{LRS} - G_{HRS}) \quad (1)$$

Let, the conductance for training = G_T , simulated synaptic weight = W , G_{LRS} = LRS conductance of CA, and G_{HRS} = HRS conductance of CA.

Regarding the weights obtained through software, they are represented using the float32 data type, offering sufficient precision for our needs. Given that the device responsible for applying conductance to the array can handle a range from 0.1A to as precise as 0.1pA, there should be confidence in the CA's conductance resolution to represent and manipulate these synaptic weights accurately.

In terms of the required precision for neural network applications, it varies based on the specific task and network architecture. However, with our approach and the inherent precision of the CA's design, we believe that the system is well-equipped to meet and

potentially exceed the precision requirements commonly associated with neural network applications.

Author action:

Following the reviewer's comments, we modified the 'Software-based neural network training' and 'Hardware implementation of MNIST-digit classification' sub-section in the 'Method' section to the sentences below. The modified main texts are shown in the revised manuscript.

“Software-based data processing and learning of CA.

To emulate the CA's input, the MNIST dataset was modified where its input was reduced to 320 units and normalized pixel values were rounded off to either 0 or 1. This process, though digital in nature, only digitized the input, not the weights or the internal operations of the CA. The device array utilizes experimentally derived weights, which were adjusted to match the conductance scale and input current scale of the array. This ensures that the CA processes genuine analog values throughout its computation.

To elaborate on the learning process in more detail: the reduced input of 320 was chunked into groups of 10, which were then used as inputs for an array with three lines. As a result, each section received 32 inputs and produced 3 outputs. The output generated by multiplying with the weights represents the score for the handwriting classification within each divided section of 10. These scores are then aggregated, and a soft-max function is applied to estimate the final input.

Hardware implementation for MNIST-digit classification

The single-layer neural network (NN) was trained using preprocessed MNIST digits with a software-based training method. The weight values of the trained network were derived by

mapping analog values between binary 0 and 1. These resulting weights for each cell in the Crossbar Array (CA) were analog values ranging between 0 and 1. Subsequently, these weights were converted to corresponding conductance values within the reading current range of the SRM, specifically between 0.1 and 1.5 nA. In this conversion, a value of "0" from the software-based procedure was mapped to a High-Resistance State (HRS), while a value of "1" was mapped to a Low-Resistance State (LRS). Intermediate values between 0 and 1 were proportionally converted to intermediate conductance states between HRS and LRS following Eq. (2) in the 'Discussion' section.

Our digit classification procedure, designed for the CA array, involves dividing the input data into ten distinctive chunks, given our input size of 320. This segmentation is crucial for our architecture, which relies on processing individual data chunks through dedicated modules to leverage the CA array's intrinsic parallel processing capabilities. Each module is assigned the task of capturing unique features from its respective segment of the data. After individual processing, the outputs from all these modules are aggregated, resulting in a combined prediction. This combined prediction is then transformed to provide probability scores for the digits 0, 1, and 2.

Our network architecture is uniquely designed for this data segmentation. Instead of a traditional fully connected approach, we separate the input data into ten smaller chunks, each comprising 32 units. Each of these chunks is processed through a specific module, consisting of a linear layer with 32 input features and 3 output features. The collective output from these ten modules produces the final prediction.”

Q6. In Figure 4, it would be insightful to understand if there are any differences between the devices stuck at High Resistance State (HRS) and Low Resistance State (LRS) for the defective devices.

Answer:

We appreciate the reviewer for the critical comment. Following the reviewer's comment, we conducted a detailed investigation into the impact of 'LRS stuck defects' in the cellular automaton (CA) concerning the accuracy of classification tasks. To explore the effects of these defects on the classification process, we employed the D30 configuration, featuring a 30% defect rate, specifically comprised of 'LRS stuck cells'. These cells in the stuck state were achieved through programming (training) to attain the conductance of LRS. During the programming process, we adopted the 1/3 bias scheme. The conductance mapping results are presented in Fig. R4a.

To assess the accuracy of training, the VMM operations were conducted. The results of VMM operations were then compared with the calculated VMM results of each column, as illustrated in Figure R4b. Remarkably, no significant differences between the two parameters were observed, indicating accurate VMM operations despite the presence of 'LRS stuck defect cells'.

Utilizing the 'D30 (LRS)' CA configuration, we conducted classification operations using the MNIST dataset. The representative results of these operations are depicted in Fig. R4c. In the representative results from the D30 (LRS) CA, misclassifications for digits 0 and 1 were observed. Additionally, the statistical results for classification using this CA are presented in Figure R4d. In comparison to the D30 CA, significant degradations in accuracy were noted for each digit. Specifically, the classification accuracies for digits 0, 1, and 2 were 67%, 37%, and 33%, respectively. Furthermore, the overall accuracy of the CA with D30 (LRS) defects

was 44.5%, as shown in Figure R6e. These results reveal that the impact of degradation for the classification accuracy which is constructed with LRS stuck defects is more critical than HRS stuck defects. These results highlight that the impact of degradation on classification accuracy, when constructed with 'LRS stuck defects,' is more critical than that of 'HRS stuck defects.'

Figure R4. Hardware-based classification results using D30 (LRS) of CA

Author action:

In response to the reviewer's feedback, we have incorporated the classification results obtained using D30 (LRS) into Supplementary Figure S6. Additionally, we have included the following commentary in the revised manuscript on page

"Furthermore, the impact of defects, specifically those involving 'LRS stuck cells', was thoroughly investigated, as illustrated in Supplementary Figure S6."

Supplementary Fig. S6: Fully hardware-based demonstration of a single-layer neural network for MNIST data classification utilizing LRS stuck defect rate of 30% (D30, LRS). **a** Trained weight-mapping results of the D30 (LRS) CA. After the training process, all cells in the CA were read out at a 2 V reading voltage. **b** VMM operation results of the D30 (LRS) CA. VMM operation results (red bars), trained weight summations (green circles) are compared to demonstrate the discrepancy between the VMM operation results and calculated values which indicates the feasibility of the VMM operation. **c** The representative experimental demonstration of the fully hardware-based classification of the MNIST data using the D30 (LRS) CA. The classification result is indicated by the blue bars, which represent the maximum values of the sensed signals. **d** Classification accuracy for each digit based on 6 classification. **e** Classification accuracies for each digit in different defective CAs and the total classification accuracy. The total classification accuracy of D30 (LRS) CA was 44.5%, which are lower than that of D30 CA. These results reveal that the impact of degradation for the classification accuracy which is constructed with LRS stuck defects is more critical than HRS stuck defects. These results highlight that the impact of degradation on classification accuracy, when constructed with 'LRS stuck defects,' is more critical than that of 'HRS stuck defects.'

Q7. For Figure 5, considering that both conditions at a reading margin of 50 yielded 100% accuracy at D0 configuration, it would be more conclusive to compare the performance at D30 or D50 configurations. Alternatively, investigating performance at a lower (< 15) reading margin would also provide valuable insights.

Answer:

We appreciate the positive feedback. In response to the reviewer's suggestions, we conducted classification based on the defect rate of the CA under the read margin condition of 50, as depicted in Fig. R5. Fig. R5a illustrates the conductance mapping for a defect rate of 30% (D30) of the CA, while Fig. R5b presents classification results for each digit. Similarly, utilizing a defect rate of 50% (D50) for the CA, we have provided the conductance mapping result (Fig. R5c), and the classification accuracy for each digit (Fig. R5d). Consequently, the overall classification accuracies are summarized in Fig. R5e.

The CA with defect rate D0 achieved a classification accuracy of 100%, while the CAs of D30 and D50 achieved classification accuracies of 69.3% and 51.6%, respectively. From this experiment, two main conclusions were drawn. Firstly, as demonstrated earlier in Fig. 4 of the manuscript, the results showed a decrease in classification accuracy with increasing defect rates. Secondly, when compared to the results under the read margin of 15 conditions, an increase in accuracy of 0.8% and 1.8% was observed in the D30 and D50 systems, respectively. Although there was an increase in accuracy, the magnitudes were not significant, and therefore, no clear and distinct differences based on the read margin could be observed.

As the reviewer correctly points out, the classification results under a read margin of 50 are more conclusive. However, what the authors want to emphasize in this study is that, even with a low read margin, the non-filamentary switching device can achieve high accuracy due to its uniform operating characteristics. This aspect highlights its strength as a component in a neural network hardware accelerator. From this perspective, results obtained under

conditions with a low read margin might be more compelling. Therefore, we consider adding these results as a Supplementary Figure to aid readers' understanding. We sincerely appreciate the reviewer's earnest feedback, which contributes to the overall quality of our research.

Fig. R5. Hardware-based classification results using D30 and D50 in the condition of $\times 50$ read margin. **a** illustrates the conductance mapping for a defect rate of 30% (D30) of the CA, while **b** presents classification results for each digit. Similarly, **c** utilizing a defect rate of 50%

(D50) for the CA, we have provided the conductance mapping result, and **d** the classification accuracy for each digit. **e** The overall classification accuracies are summarized. The CA with defect rate D0 achieved a classification accuracy of 100%, while the CAs of D30 and D50 achieved classification accuracies of 69.3% and 51.6%, respectively.

Author action:

We added the classification results using D30 and D50 in the condition of x 50 read margin in the Supplementary Fig. 7 in the revised ‘Supplementary Information’ like below.

Supplementary Fig. S7: Hardware-based classification results using D30 and D50 in the condition of x 50 read margin. a illustrates the conductance mapping for a defect rate of 30% (D30) of the CA, while b presents classification results for each digit. Similarly, c utilizing a defect rate of 50% (D50) for the CA, we have provided the conductance mapping result, and d the classification accuracy for each digit. e The overall classification accuracies are summarized. The CA with defect rate D0 achieved a classification accuracy of 100%, while

the CAs of D30 and D50 achieved classification accuracies of 69.3% and 51.6%, respectively. From this experiment, two main conclusions were drawn. Firstly, as demonstrated earlier in Fig 4 of the manuscript, the results showed a decrease in classification accuracy with increasing defect rate. Secondly, when compared to the results under the read margin of 15 conditions, an increase in accuracy of 0.8% and 1.8% was observed for D30 and D50 systems, respectively. Although there was an increase in accuracy, the magnitudes were not significant, and therefore, no clear and distinct differences based on the read margin could be observed.

And, we added the brief commentary on the ‘Impact of the SRM Reading Margin on Classification Accuracy’ in the revised manuscript as below.

“In addition, to compare with the smaller read margin condition of CA, the classification results using D30 and D50 for x 50 read margin condition are demonstrated in Supplementary Fig. S7.”

Q8. The selection functionality is pivotal to a selector-free CA. It is essential to elaborate if there are any specific requirements regarding the level of selectivity the self-rectifying property needs to provide for decent performance.

Answer:

We sincerely appreciate your comments. As the reviewer is aware, high selectivity is an essential characteristic required for achieving high array density, and numerous studies related to the required selectivity for specific densities have been conducted by various research institutions. [R4, R5] In Ref R4, numerical calculations demonstrated that a device with a selectivity of 3.3×10^2 could be applied to a maximum CA size of 6 kb. Therefore, it can be anticipated that the selectivity required for application to a 1 kb array would be lower than this value. Additionally, in ref. R5, the investigation of the necessary selectivity for the selector in a 1S1R device indicates a range between 10 to 10^2 for integration into a 1 kb CA. Based on these studies, it has been reported that determining the precise numerical values for the minimum selectivity required for a device applicable to an array with a density of 1 kb is challenging. However, estimates suggest that it lies in the range of approximately 10 to 10^2 orders of selectivity. Therefore, the selectivity of about 10^4 for the developed SRM device can be deemed sufficient for its application in a 1 kb array.

Furthermore, in our previous research, we demonstrated the application of this SRM device in arrays of up to 100 kb through experimental results. According to the research findings, the SRM device exhibited characteristics unaffected by sneak current up to 25 kb arrays. However, in 100 kb arrays, some influence from sneak current was observed. However, through an appropriate biasing scheme, it was confirmed that sneak current can be effectively suppressed, enabling its application in larger arrays.

Consequently, the selectivity of our proposed SRM integrated into 1 kb CA is

sufficiently high value. Furthermore, through the experimental investigation, the SRM can be integrated into 100 kb CA without significant deterioration, ensuring robust performance using the SRM-integrated 1 kb CA. Once again, we fully agree with the reviewer that selectivity is an extremely important factor when applying high-density arrays.

Author action:

We investigated research demonstrating numerical study of the required selectivity for the specific array densities shown below,

R4. Huang, J. J et al. One selector-one resistor (1S1R) crossbar array for high-density flexible memory applications. *2011 International Electron Devices Meeting (IEDM)*. (2011)

R5. Huang, C. -H et al. Self-Selecting Resistive Switching Scheme Using TiO₂ Nanorod Arrays. *Sci. Rep.*7, 2066 (2017)

Q9. In addition to the reading margin and selection, exploring the impact of Cell-to-Cell (C2C) and Device-to-Device (D2D) variations on CA performance is crucial, as both are fundamental characteristics for memristor devices.

Answer:

Thanks for the reviewer's sincere comment. As the reviewer pointed out, we completely agree that the variability in the operation of array devices directly impacts their performance. Therefore, in our study, we have qualitatively presented the read current variability of the 1 kb array device incorporating SRM. Although this information was mentioned in the manuscript, we acknowledge that presenting this data in a figure would enhance readability. We will make the necessary modifications by including this numerical data in a figure. Once again, we sincerely appreciate the reviewer's valuable comments.

Author action:

We add the coefficient of variation values calculated in Fig. 2c, Fig 2f, Fig 5c, Fig 6a, and Fig 6b in the revised manuscript. The revised Figures are shown below.

Fig. 2: Electrical characteristics of our 1 kb CA device. **a** Schematic of the CA utilizing a one-third biasing scheme during the DC $I-V$ characterization. **b** DC $I-V$ characteristics of 1024 cells in the CA. The black curve marked ① denotes the SET process, while the black curve marked ② denotes the RESET process. **c** Reading current distributions of each state. The coefficient of variations of the LRS and HRS were 0.057 and 0.06, respectively. **d** Schematic of the AC-based operations utilizing the biasing scheme. Inhibiting voltages were induced in the unselected cells and had a larger pulse width than the operational voltage **e** Electrical pulse-induced behavior of 30 cells in the CA. **f** Amplitude distributions for the SET and RESET processes. Very low deviation values were observed in each process. The highly uniform electrical characteristics of the CA can be attributed to the nonfilamentary characteristics of our previously developed SRM device.

Fig. 5: Investigation of the impact of the read margin on classification accuracy. a $DC I-V$ characteristics of the SRM-integrated CA, and **b** retention characteristics measured at a reading voltage of 1.7 V. Robust nonvolatility was observed at 1.7 V. **c** Read current distributions of the LRS and HRS at 1.7 V. The coefficient of variations of the HRS and LRS (0.093 and 0.142, respectively) were relatively low. **d** Trained weight-mapping results obtained at a reading voltage of 1.7 V. **e** Comparison of the VMM operation results and calculated weight summations in each column. The feasibility of VMM operations using a 1.7 V reading voltage was confirmed. **f** Representative classification results of the CA under a 1.7 V reading

voltage, and **g** total classification accuracy for each digit. All digits were classified accurately (1500 classification operations). Comparison of the two different reading margins ($\times 50$ @ 1.7 V and $\times 15$ @ 2.0 V) revealed that the reading margin of the nonfilamentary-type memristor had an insignificant effect on the classification accuracy because of its highly uniform operating characteristics.

Fig. 6: Investigation of the impacts of a non-selective memristor integrated into the passive CA on VMM operation accuracy. a DC I - V characteristics of a unit memristor device

without selection functionality (left). Ten randomly chosen cells were measured over three cycles (total number of DC $I-V$ curves: 30). The read current distributions of each state were extracted from the DC $I-V$ curves (right). **b** DC $I-V$ characteristics of an 8×8 CA integrated with the memristor. All cells in the CA were measured in one cycle in DC $I-V$ sweep mode (graph). The read current distributions of each state were extracted from the DC $I-V$ curve (right). **c** Programming scheme of the 8×8 CA. A total of 32 randomly chosen cells were programmed to the LRS (black), and the rest of the cells were maintained in the HRS (orange). **d** Resistance distribution of the 8×8 CA after the programming step. The programmed weight values of the CA were not coincident with the proposed programming scheme, indicating that programming and reading were not processed accurately owing to sneak current effects. **e** Results of VMM operations using the 8×8 CA. Nearly identical output currents were observed in each column, which implies that the VMM operation was not processed accurately because of significant interferences between integrated cells in the CA.

R2

This paper presents a Crossbar Array (CA) based on self-rectifying memristors as hardware accelerators for artificial neural network (ANN) applications. A 1 kb CA is implemented as a hardware accelerator for neural network operations, and the impact of various CA characteristics on image classification tasks is investigated, achieving 100% accuracy in a three-class classification on the MNIST dataset. However, there are some concerns that need to be addressed to enhance the paper's value and clarity.

Answer: We sincerely appreciate the reviewer's thoughtful comments. We did our best to address the concerns raised by the reviewer.

Q1. This paper could benefit from a comparative analysis with other similar research in the field. Including a table or graph comparing the performance of the proposed CA with other hardware accelerators for ANNs would help readers understand the significance and value of this work. Such a comparison should be highlighted in the abstract and conclusion sections to emphasize the paper's contributions.

Answer:

We appreciate the reviewer's helpful comment. In response to the reviewer's feedback, we conducted an investigation into other research focusing on neural network applications utilizing memristor crossbar array-based hardware accelerators. Most of the studies in this area utilized the 1T1R (1-transistor and 1-resistor) configuration, showcasing outstanding performances in terms of high array yield, energy efficiency, and classification accuracy, as demonstrated in the table below. Additionally, there have been neural network applications employing passive crossbar arrays, specifically in the 1S1R (1-selector and 1-resistor) configuration, for simpler classification tasks. These studies achieved highly accurate classification results, especially for datasets like the 'alphabet,' although the applied crossbar array density was limited to 9 by 9. However, it is worth noting that fully hardware-based neural network applications utilizing Self-Rectifying Crossbar (SRM)-based crossbar arrays have not been widely reported; most studies have relied on simulation-based results. These limitations often stem from the insufficient reliability of the devices and arrays.

In our study, we have addressed these challenges by demonstrating the fully hardware-based classification of MNIST data with high accuracy using a purely SRM-based passive crossbar array. While we acknowledge the limited number of test runs, the constrained test data, and the relatively modest energy efficiency demonstrated in our study, we emphasize that this

research represents the first case where such performance has been demonstrated using an SRM-based passive crossbar array. We hope that this pioneering effort will pave the way for future advancements in the field. Thank you for highlighting the significance of this achievement.

	This work	Ref. 1	Ref. 2	Ref. 3	Ref. 4	Ref. 5
Array configuration	1R (SRC)	1S-1R	1T-1R	1T-1R	1T-1R	1T-1R
Array density	32 × 32 (1 kb)	9 × 9 (< 1 kb)	128 × 64 (8 kb)	54 × 108 (~ 5 kb)	128 × 16 (2 kb)	128 × 64 (8 kb)
Array yield (%)	100	100	98.9	Not available	99	Not available
Energy efficiency (TOPS/W)	4.35	Not available	115	1.37	11	77.4
Application	MNIST classification	Pattern classification	MNIST classification	Pattern classification	MNIST classification	MNIST classification
Accuracy	100	100	89.9	94.6	96.19	92.3
Integrated device stack	Ru/HfSiO _y /Al ₂ O ₃ /HfSiO _x /TiN	Pt/Ru/TiO ₂ /RuO ₂ /Pt/HfO ₂ /TiN	Ta/HfO _x /Pd	Pd/WO _x /Au	TiN/TaO _x /HfO _x /TiN	Pt/TaO _x /Ta

Table R1. The comparison results of hardware implementation for neural network applications using memristive crossbar arrays

Author action:

Following the reviewer's comments, we have created a table comparing the results of various hardware-based neural network applications. The table includes comparisons based on

array configuration, array yield, application, accuracy, and energy efficiency. And, we added the comparing table in the ‘Conclusion’ section in the revised manuscript with a brief commentary as below,

“Through this study, the neural network applications utilizing memristive crossbar arrays are compared in Table 1. It is noteworthy that our study represents the first reported demonstration in the field of SRMs, utilizing a purely self-rectifying memristor-based passive crossbar array for hardware implementations of neural network applications. While the energy efficiency of the proposed Crossbar Array (CA) is currently lower than Ref. 36 and 55, we anticipate that higher efficiency can be achieved by increasing the array density.”

	This work	Ref. 1	Ref. 2	Ref. 3	Ref. 4	Ref. 5
Array configuration	1R (SRC)	1S-1R	1T-1R	1T-1R	1T-1R	1T-1R
Array density	32 × 32 (1 kb)	9 × 9 (< 1 kb)	128 × 64 (8 kb)	54 × 108 (~ 5 kb)	128 × 16 (2 kb)	128 × 64 (8 kb)
Array yield (%)	100	100	98.9	Not available	99	Not available
Energy efficiency (TOPS/W)	4.35	Not available	115	1.37	11	77.4
Application	MNIST classification	Pattern classification	MNIST classification	Pattern classification	MNIST classification	MNIST classification
Accuracy	100	100	89.9	94.6	96.19	92.3
Integrated device stack	Ru/HfSiO_y/Al₂O₃/HfSiO_x/TiN	Pt/Ru/TiO₂/RuO₂/Pt/HfO₂/TiN	Ta/HfO_x/Pd	Pd/WO_x/Au	TiN/TaO_x/HfO_x/TiN	Pt/TaO_x/Ta

Table 1. The comparison results of hardware implementation for neural network applications using memristive crossbar arrays.

Q2. This paper claims that the memristor-based passive crossbar array can accelerate neural network operations, but it lacks specific chip performance metrics, such as computational speed and power consumption. Providing these details and comparing the CA's performance with traditional microelectronic chips like CPUs, GPUs, and ASICs would demonstrate its advantages more clearly.

Answer:

We thank for the reviewer's highly insightful comment. We conducted the numerical calculation of the energy efficiency for the VMM operation using the proposed CA. The applied equation for the calculating energy efficiency of the VMM performance is shown below.

Tera operations per second per Watt (TOPS/W) = $(2 \times N \text{ operations}) / [(Read \text{ latency}) \times (Power \text{ consumption})]$

In our study, we utilized the parameters $N = 32$, Read latency = $10 \mu\text{s}$, and Power consumption = $1.47 \mu\text{W}$ for our calculations, resulting in an energy efficiency of 4.35 TOPS/W for the Vector-Matrix Multiplication (VMM) performance. As evident from the table, the energy efficiency of our system was relatively lower compared to other memristor crossbar array-based hardware accelerators. There are two main reasons for this relatively lower energy efficiency. Firstly, the array size used for the performance evaluation was 1 kb, indicating a relatively lower density, which affected the energy efficiency. Secondly, the integrated memristors operated with a latency of $10 \mu\text{s}$, impacting the overall energy consumption.

To address these limitations and enhance energy efficiency, our ongoing research focuses on improving the array's density by increasing integration and optimizing memristor latency. We anticipate that by enhancing the integration density of the array and optimizing memristor latency, we can achieve higher energy efficiency in future iterations of our system. We are actively working on these improvements, and we plan to provide detailed demonstrations in the subsequent phases of our research. Furthermore, we compared the energy efficiency of the proposed CA with highly matured and traditional microchips such as FPGAs and ASICs, shown below. Once again, we express our gratitude for the helpful comments provided, which have guided our ongoing research efforts.

	This work	Ref. 56	Ref. 57	Ref. 58	Ref. 59	Ref. 60	Ref. 61
Hardware	Memristive passive crossbar array	GPU (Titan X)	FPGA (Virtex 7)	FPGA (Arria-10)	FPGA (Zynq-7000)	ASICs (YodaNN)	ASICs (Google TPU)
Energy efficiency	4.35 TOPS/W	98 GOPS/W	935 GOPS/W	30 GOPS/W	14.3 GOPS/W	8.6 TOPS/W	2.3 TOPS/W

Table R2. The comparison results with traditional microelectronic chips, which function as hardware accelerators for computation.

Author action:

Following the reviewer’s comment, we added the table comparing results and brief commentary in the ‘Conclusion’ section in the revised manuscript as below,

“Furthermore, we compare the energy efficiency of our hardware accelerator with traditional microelectronic chips such as GPUs, FPGAs, and ASICs in Table 2. The results of the comparison show that our hardware accelerator achieves relatively high energy efficiency due to significantly lower power consumption compared to traditional microelectronic chips. For

detailed calculations, refer to Supplementary Note 1.”

	This work	Ref. 56	Ref. 57	Ref. 58	Ref. 59	Ref. 60	Ref. 61
Hardware	Memristive passive crossbar array	GPU (Titan X)	FPGA (Virtex 7)	FPGA (Arria-10)	FPGA (Zynq-7000)	ASICs (YodaNN)	ASICs (Google TPU)
Energy efficiency	4.35 TOPS/W	98 GOPS/W	935 GOPS/W	30 GOPS/W	14.3 GOPS/W	8.6 TOPS/W	2.3 TOPS/W

Table 2. The comparison results with traditional microelectronic chips, which function as hardware accelerators for computation.

Additionally, we included detailed information on the calculation process of energy efficiency in Supplementary Note 1, which is provided in the revised 'Supplementary Information'. The added text is shown in below.

“Supplementary Note 1: The calculation process of energy efficiency of proposed 1 kb passive CA for implementing vector-matrix multiplication (VMM) operation.

To assess the energy efficiency of our proposed 1 kb passive Crossbar Array (CA) for the hardware-based implementation of Vector-Matrix Multiplication (VMM) operations, we calculated the 'Tera operations per second per Watt (TOPS/W)' using the equation:

$$\frac{TOPS}{W} = \frac{2 \times N \text{ operations}}{\text{Read latency} \times \text{Power consumption}}$$

In our study, we employed the parameters N=32, Read latency = 10 μs, and Power consumption = 1.47 μW, extracted from the defect-free (D0) CA, for our calculations. This yielded an energy efficiency of 4.35 TOPS/W for the VMM performance.”

Q3. This paper mentions additional power consumption due to the application of different voltages to suppress sneak currents. It would be helpful to specify the proportion of this

additional power consumption relative to the total power consumption and its impact on the overall chip performance. This information would enable readers to better evaluate the trade-offs involved in using the proposed CA for ANN applications.

Answer:

We are grateful for the reviewer's insightful comment. As mentioned, we implemented different voltages, referred to as 'inhibit biases,' to the word and bit lines shared by 'unselected cells' to suppress sneak currents. Selecting an appropriate inhibit biasing scheme is pivotal for the operation of crossbar array devices, as it not only affects the suppression of sneak currents but also additional power consumption, as the reviewer is aware.

To ensure the application of an appropriate inhibit biasing scheme, we conducted a numerical analysis to assess the limits of array density and power consumption associated with the scheme. The results, depicted in Figure R6, were obtained under the worst-case scenario, assuming that all unselected cells in the array were in the Low-Resistance State (LRS). When evaluating the limit of array density through read margin, we found that with a critical read margin set to 10% and a 1/3 biasing scheme applied, the density corresponding to approximately 200 kb could be achieved with 412-word lines. In the case of the 1/2 scheme, it was predicted that arrays of about 19 kb could be applied with 140-word lines.

Furthermore, when calculating the additional power consumption resulting from the biasing scheme, we observed the lowest increase in power consumption with the increase in array size when the 1/3 scheme was applied, consistent with the read margin results mentioned earlier. These numerical analyses provided the foundation for our study's approach. The evaluation results obtained from these analyses were validated by applying them to the actual array through a previous study. Consequently, in this study, we implemented the 1/3 biasing

scheme based on the results gained from these analyses and the validation process.

	R_{HRS} (Ω)	R_{LRS} (Ω)	R_{s1} (Ω)	R_{s2} (Ω)	R_{s3} (Ω)	R_{pu} (Ω)	N
1/2 scheme	4.00E+09	1.00E+09	2.50E+10	1.00E+13	2.50E+10	1.00E+09	32
1/3 scheme	4.00E+09	1.00E+09	1.75E+11	4.67E+12	1.75E+11	1.00E+09	32

Fig. R6: The numerical analyses of 1 kb CA under different biasing schemes. To investigate the influence of biasing schemes for 1 kb CA, two different bias schemes were applied. **a** The schematic figure of 1 kb CA under biasing schemes. Under the biasing schemes, the selected cell is connected the bit line biased to V_{op} and word line grounded. However, the unselected cells are partitioned into three groups, which are referred as R_{s1} , R_{s2} , and R_{s3} . The group of R_{s1} (green colored cells) are connected bit line biased to V_{op} and word lined biased to inhibit voltage 1 (V_{in1}). The group R_{s2} (black colored cells) are connected bit line biased to inhibit voltage 2 (V_{in2}) and word lined biased V_{in1} . The group R_{s3} (blue colored cells) are connected bit line

biased to V_{in2} and word lined grounded. **b** The equivalent circuit of the CA under biasing schemes. **c** The calculation results of read margin according to each biasing scheme. **d** The calculation results of the additional power consumption stemming from the unselected cells under each biasing scheme. **e** The parameters of proposed CA for numerical calculations.

Author action:

Following the reviewer’s comment, we add the numerical analyses for each biasing scheme in Supplementary Fig. S2 in the revised ‘Supplementary Information’.

	$R_{HRS} (\Omega)$	$R_{LRS} (\Omega)$	$R_{s1} (\Omega)$	$R_{s2} (\Omega)$	$R_{s3} (\Omega)$	$R_{pu} (\Omega)$	N
1/2 scheme	4.00E+09	1.00E+09	2.50E+10	1.00E+13	2.50E+10	1.00E+09	32
1/3 scheme	4.00E+09	1.00E+09	1.75E+11	4.67E+12	1.75E+11	1.00E+09	32

Supplementary Fig. S2: The numerical analyses of 1 kb CA under different biasing schemes.

To investigate the influence of biasing schemes for 1 kb CA, two different bias schemes were applied. a The schematic figure of 1 kb CA under biasing schemes. Under the biasing schemes, the selected cell is connected the bit line biased to V_{op} and word line grounded. However, the unselected cells are partitioned into three groups, which are referred as R_{s1} , R_{s2} , and R_{s3} . The group of R_{s1} (green colored cells) are connected bit line biased to V_{op} and word lined biased to inhibit voltage 1 (V_{in1}). The group R_{s2} (black colored cells) are connected bit line biased to inhibit voltage 2 (V_{in2}) and word lined biased V_{in1} . The group R_{s3} (blue colored cells) are connected bit line biased to V_{in2} and word lined grounded. **b** The equivalent circuit of the CA under biasing schemes. **c** The calculation results of read margin according to each biasing scheme. **d** The calculation results of the additional power consumption stemming from the unselected cells under each biasing scheme. **e** The parameters of proposed CA for numerical calculations.

To calculate the read margin under each biasing scheme, the equations (1) to (3) are utilized.

$$\text{Read margin(\%)} = \frac{R_{pu}}{R_{LRS,Read} + R_{pu}} - \frac{R_{pu}}{R_{HRS,Read} + R_{pu}} \quad (1)$$

$$R_{HRS,Read} = \frac{R_{HRS} \times \left(\frac{R_{s1}}{N-1} + \frac{R_{s2}}{(N-1)^2} + \frac{R_{s3}}{N-1} \right)}{R_{HRS} + \left(\frac{R_{s1}}{N-1} + \frac{R_{s2}}{(N-1)^2} + \frac{R_{s3}}{N-1} \right)} \quad (2)$$

$$R_{LRS,Read} = \frac{R_{LRS} \times \left(\frac{R_{s1}}{N-1} + \frac{R_{s2}}{(N-1)^2} + \frac{R_{s3}}{N-1} \right)}{R_{LRS} + \left(\frac{R_{s1}}{N-1} + \frac{R_{s2}}{(N-1)^2} + \frac{R_{s3}}{N-1} \right)} \quad (3)$$

In addition, we added the sentence below on page 7 of the revised manuscript.

“Furthermore, the numerical analyses for the limited density of the array and additional power

consumption stemming from the sneak current under applying each biasing scheme were conducted (Supplementary Fig. S2)."

Q4. This paper should provide clearer definitions of the terms "Experimental result," "Calculated result," and "Simulated result." Additionally, quantitative descriptions of how well the experimental and calculated results match, in terms of bit precision, would help readers better understand the accuracy of the proposed CA. The paper should also explain the required bit precision for neural network applications and demonstrate that the CA can meet these requirements.

Answer:

Thank you for the insightful comment. The 'Experimental results', 'Calculated results', and 'Simulated results' indicate the output current sensed (experimentally measured) from the VMM operations, the sum of read current (conductance) in programmed CA (numerical calculation of current sum at each column based on the programmed (already known) conductance states), and the sum of conductance trained through simulation (the required VMM result to achieve the inference accuracy of 100%), respectively. All these parameters represent values sequentially sensed and calculated for each column. In addition, it's essential to clarify the difference between the nature of the proposed CA and conventional digital architectures when discussing precision. While typical digital systems rely on bit precision, the proposed CA operates primarily on analog signals. This fundamental difference means that our approach does not directly engage with bit precision in the traditional sense. However, it's crucial to understand the steps taken to align the MNIST data set with the CA's operational mode.

Fig. R7: Software-based data processing and learning of CA for classification tasks

To emulate the CA's input, the MNIST dataset was modified where its input was reduced to 320 units and normalized pixel values were rounded off to either 0 or 1. This process, though digital in nature, only digitized the input, not the weights or the internal operations of the CA. The device array utilizes experimentally derived weights, which were adjusted to match the conductance scale and input current scale of the array. This ensures that the CA processes genuine analog values throughout its computation.

To elaborate on the learning process in more detail: the reduced input of 320 was chunked into groups of 10, which were then used as inputs for an array with three lines. As a result, each section received 32 inputs and produced 3 outputs. The output generated by multiplying with the weights represents the score for the handwriting classification within each divided section of 10. These scores are then aggregated, and a soft-max function is applied to estimate the final input.

The calculated weights were converted to the target conductance and applied to the CA using the following formula:

$$G_T = W \times (G_{LRS} - G_{HRS}) \quad (1)$$

Let, the conductance for training = G_T and simulated synaptic weight = W .

Regarding the weights obtained through software, they are represented using the float32 data type, offering sufficient precision for our needs. Given that the device responsible for applying conductance to the array can handle a range from 0.1A to as precise as 0.1pA, there should be confidence in the CA's conductance resolution to represent and manipulate these synaptic weights accurately.

In terms of the required precision for neural network applications, it varies based on the specific task and network architecture. However, with our approach and the inherent precision of the CA's design, we believe that the system is well-equipped to meet and potentially exceed the precision requirements commonly associated with neural network applications.

Author action:

Following the reviewer's comments, we added clearer definitions of the terms 'Experimental results', 'Calculated results', and 'Simulated results' on page 11 of the revised manuscript as below.

“The ‘Experimental results’, ‘Calculated results’, and ‘Simulated results’ indicate the output current sensed (experimentally measured) from the VMM operations, the sum of read current (conductance) in programmed CA (numerical calculation of current sum at each column based on the programmed (already known) conductance states), and the sum of conductance trained through simulation (the required VMM result to achieve the inference accuracy of 100%), respectively. All these parameters represent values sequentially sensed and calculated for each

column. The detail of VMM operation using CA is demonstrated in the ‘Method’ section”

Furthermore, we added the quantitative description of accurate operation in terms of bit precision and required bit precision in the ‘Discussion’ section in the revised manuscript (on page 17) as below.

“In addition, considering the classification accuracy with terms of bit precision in our system, it is essential to clarify the difference between the nature of the proposed CA conventional digital architectures. While typical digital systems rely on bit precision, the proposed CA operates primarily on analog signals. This fundamental difference means that our approach does not directly engage with bit precision in the traditional sense. However, it's crucial to understand the steps taken to align the MNIST data set with the CA's operational mode.

MNIST's input was reduced to 320 units and normalized pixel values were rounded off to 0 or 1 to simulate the CA's input. Though digital, this process merely digitized input, not weights or CA procedures. The device array utilizes experimental weights adjusted to match its conductance and input current scales. This ensures the CA computes using genuine analog values. To elaborate on the learning process, the reduced input of 320 was chunked into groups of 10 and fed to a three-line array. Each segment has 32 inputs and 3 outputs. Multiplying by weights yields the handwriting classification result for each 10-part segment. The final input is estimated using a soft-max function after aggregating these scores. The calculated weights were converted to the target conductance and applied to the CA using the following formula:

$$G_T = W \times (G_{LRS} - G_{HRS}) \quad (1)$$

Let, the conductance for training = G_T and simulated synaptic weight = W .

Regarding the weights obtained through software, they are represented using the float32

data type, offering sufficient precision for our needs. Given that the device responsible for applying conductance to the array can handle a range from 0.1A to as precise as 0.1pA, there should be confidence in the CA's conductance resolution to represent and manipulate these synaptic weights accurately.

In terms of the required precision for neural network applications, it varies based on the specific task and network architecture. However, with our approach and the inherent precision of the CA's design, we believe that the system is well-equipped to meet and potentially exceed the precision requirements commonly associated with neural network applications.”

In addition, we modified the main text of ‘Software-based neural network training’ as below in the revised manuscript.

“Software-based data processing and learning of CA.

To emulate the CA's input, the MNIST dataset was modified where its input was reduced to 320 units and normalized pixel values were rounded off to either 0 or 1. This process, though digital in nature, only digitized the input, not the weights or the internal operations of the CA. The device array utilizes experimentally derived weights, which were adjusted to match the conductance scale and input current scale of the array. This ensures that the CA processes genuine analog values throughout its computation. To elaborate on the learning process in more detail: the reduced input of 320 was chunked into groups of 10, which were then used as inputs for an array with three lines. As a result, each section received 32 inputs and produced 3 outputs. The output generated by multiplying with the weights represents the score for the handwriting classification within each divided section of 10. These scores are then aggregated, and a soft-max function is applied to estimate the final input.”

Q5. The authors should provide a more detailed description of the digit classification process using the CA array. This should include clarifying whether the network is fully connected or not, and explaining the significance of the localized "winner-takes-all" rule. Additionally, detailed information on the selection rules for the training and testing sets leading to the claimed 100% accuracy in classifying handwritten digits 0, 1, and 2 should be provided. Furthermore, the paper should explain how the multiplication of currents read from each column of the CA is achieved, including whether digital signals are used, and how this process affects overall performance.

Answer:

We sincerely appreciate the insightful feedback, particularly regarding the detailed description of our digit classification process using the CA array and other associated queries.

Our digit classification procedure, designed with the CA array in mind, divides the input data into ten distinctive chunks, given our input size of 320. This segmentation is essential for our architecture, which relies on processing individual data chunks through dedicated modules to leverage the CA array's intrinsic parallel processing capabilities. Each module is tasked with capturing unique features from its respective segment of the data. After this individual processing, the outputs from all these modules are aggregated, leading to a combined prediction. This combined prediction is subsequently transformed to provide probability scores for the digits 0, 1, and 2.

Our network architecture is designed uniquely for this data segmentation. Instead of a traditional fully connected approach, we separate the input data into these ten smaller chunks, each of 32 units. Each of these chunks is then processed through a specific module, consisting

of a linear layer with 32 input features and 3 output features. The collective output from these ten modules gives the final prediction.

While our methodology does not use the conventional "winner-takes-all" mechanism, our design still upholds a principle reminiscent of its philosophy. Once the outputs from all the modules are combined, the subsequent Soft-max transformation is applied. This transformation effectively squashes the combined output into a probability distribution over the predicted output classes. Within this distribution, the neuron (or class) with the highest original activation will have a significantly higher probability. This resembles a soft version of the "winner-takes-all" strategy, where instead of making an absolute decision, the system provides a probability score, emphasizing the most likely class while still retaining information about other classes.

Concerning our dataset, we simply selected the instances of digits 0, 1, and 2 from the MNIST dataset. The test accuracy happened to be 100% only within the test samples out of the selected dataset but it may converge into the overall accuracy of 98.8% if more test data was investigated shown in Fig. R8.

Fig. R8. Software-based classification accuracy for training and inferencing (test).

Author action:

We have revised the main text in the 'Hardware implementation for MNIST-digit classification' within the 'Method' section as below.

“The single-layer neural network (NN) was trained using preprocessed MNIST digits with a software-based training method. The weight values of the trained network were derived by mapping analog values between binary 0 and 1. These resulting weights for each cell in the CA were analog values ranging between 0 and 1. Subsequently, these weights were converted to corresponding conductance values within the reading current range of the SRM, specifically between 0.1 and 1.5 nA. In this conversion, a value of "0" from the software-based procedure

was mapped to a High-Resistance State (HRS), while a value of "1" was mapped to a Low-Resistance State (LRS). Intermediate values between 0 and 1 were proportionally converted to intermediate conductance states between HRS and LRS following Eq. (2) in the 'Discussion' section.

Our digit classification procedure, designed for the CA array, involves dividing the input data into ten distinctive chunks, given our input size of 320. This segmentation is crucial for our architecture, which relies on processing individual data chunks through dedicated modules to leverage the CA array's intrinsic parallel processing capabilities. Each module is assigned the task of capturing unique features from its respective segment of the data. After individual processing, the outputs from all these modules are aggregated, resulting in a combined prediction. This combined prediction is then transformed to provide probability scores for the digits 0, 1, and 2.

Our network architecture is uniquely designed for this data segmentation. Instead of a traditional fully connected approach, we separate the input data into ten smaller chunks, each comprising 32 units. Each of these chunks is processed through a specific module, consisting of a linear layer with 32 input features and 3 output features. The collective output from these ten modules produces the final prediction.”

Additionally, we changed the term 'winner-takes-all' to 'max-current sensing' to enhance clarity regarding the method used for determining the classification results.

REVIEWERS' COMMENTS

Reviewer #1 (Remarks to the Author):

The authors have successfully addressed all my concerns over the the previous version and made significant improvements on the quality of the manuscript. I believe it has now met the standard of Nature Communication and therefore recommend for it to be published in its current form.

Reviewer #2 (Remarks to the Author):

The authors have addressed all my concerns and technical inquiries. There is only a minor aspect that requires refinement: the guidelines for selecting training and testing sets should be clarified. Specifically, it is essential to ascertain whether all the digits (0, 1, and 2) from the MNIST dataset are employed for both training and testing. Additionally, it would be beneficial to specify the exact number of images designated for the training set and testing set, respectively.

Reviewer #1 (Remarks to the Author):

The authors have successfully addressed all my concerns over the the previous version and made significant improvements on the quality of the manuscript. I believe it has now met the standard of Nature Communication and therefore recommend for it to be published in its current form.

Answer:

Thank for the reviewer's thoughtful and constructive feedback on our paper. We greatly appreciate the time and effort the reviewer dedicated to reviewing our work. The reviewer's insightful comments have been invaluable in enhancing the quality and depth of our work.

Reviewer #2 (Remarks to the Author):

The authors have addressed all my concerns and technical inquiries. There is only a minor aspect that requires refinement: the guidelines for selecting training and testing sets should be clarified. Specifically, it is essential to ascertain whether all the digits (0, 1, and 2) from the MNIST dataset are employed for both training and testing. Additionally, it would be beneficial to specify the exact number of images designated for the training set and testing set, respectively.

Answer:

Thank you for the reviewer's helpful comments. In this study, we employed the MNIST dataset consisting of 18623 samples for training and 3147 samples for testing our single-layer neural network in the 'software-based' classification. Subsequently, in the 'fully hardware-based' classification process, we evaluated the classification performance of our developed memristor CA under varying defect ratios and read margins. This assessment utilized a total of 1500 test data points, with 500 data points allocated

to each digit.

Author action:

We have added the following sentence into the ‘Software-based data processing and learning of CA’ within the ‘Method’ section in the revised manuscript.

“In the training (learning) and testing (inference) tasks in the software-based classification process, we utilized the MNIST dataset comprising 18623 and 3147 samples for training and testing our single-layer NN, respectively. Subsequently, in the fully hardware-based classification process, we evaluated the classification performance of our developed memristor CA under varying defect ratios and read margins. This assessment utilized a total of 1500 test data points, with 500 data points allocated to each digit.”